# shinyDepMap, a tool to identify targetable cancer genes and their functional connections from Cancer Dependency Map data

Kenichi Shimada*, John A Bachman, Jeremy L Muhlich, Timothy J Mitchison

Laboratory of Systems Pharmacology and Department of Systems Biology, Harvard Medical School, Boston, United States

**Abstract** Individual cancers rely on distinct essential genes for their survival. The Cancer Dependency Map (DepMap) is an ongoing project to uncover these gene dependencies in hundreds of cancer cell lines. To make this drug discovery resource more accessible to the scientific community, we built an easy-to-use browser, shinyDepMap (https://labsyspharm.shinyapps.io/depmap). shinyDepMap combines CRISPR and shRNA data to determine, for each gene, the growth reduction caused by knockout/knockdown and the selectivity of this effect across cell lines. The tool also clusters genes with similar dependencies, revealing functional relationships. shinyDepMap can be used to (1) predict the efficacy and selectivity of drugs targeting particular genes; (2) identify maximally sensitive cell lines for testing a drug; (3) target hop, that is, navigate from an undruggable protein with the desired selectivity profile, such as an activated oncogene, to more druggable targets with a similar profile; and (4) identify novel pathways driving cancer cell growth and survival.

*For correspondence:
kenichi_shimada@hms.harvard.edu

## Introduction

Cancer is a disease of the genome. Hundreds, if not thousands, of driver mutations cause cancer in different patients (*MC3 Working Group et al., 2018*), and extensive collaborative efforts such as the Cancer Genome Atlas Program (TCGA) have helped discover them (*The Cancer Genome Atlas Research Network, 2019*). Targeted therapies, a type of precision medicine, aim to treat cancer by selectively killing cancer cells with a specific genotype and spectrum of driver mutations (*Friedman et al., 2015*). The underlying hypothesis is that cancers depend on essential genes that are not the same for all tissues, and that these conditionally essential genes constitute a druggable dependency—an 'Achilles' heel'—that can be exploited to develop targeted drugs with minimal toxicity. To achieve this goal, it is important to identify conditionally essential genes for all cancers. It is also important to group these conditionally essential genes into functionally related sets to maximize the chance of finding a druggable target within each set, such as a kinase or other enzyme.

The concept of essential vs. non-essential genes arose largely from genetic research in model organisms. Traditionally, it was considered a binary distinction that held across any genotype. However, loss of a given gene can decrease cell growth without killing the cell, so it is more realistic to assign a numerical value to the degree of essentiality, that is, the extent to which loss of a gene, or inhibition of its product, influences fitness. In cancer, this value may depend on the genotype, transcriptome, and lineage of the cell. In principle, genes that are only essential in a few cell types might make better drug targets since inhibiting their function is less likely to cause toxicity in non-cancer tissues. For example, the epidermal growth factor receptor is strongly required in certain cancer

cells, but not in normal bone marrow stem cells, making it a potentially good target (*Wang et al., 2006*).

The Cancer Dependency Map (DepMap) is an ongoing project to identify essential genes across hundreds of cancer cell lines using genome-wide CRISPR and shRNA screens (*Tsherniak et al., 2017*; *Behan et al., 2019*). It has already been used successfully to discover cancer cells' genetic vulnerabilities (*Sandoval et al., 2018*; *Wang et al., 2019b*). These data represent a gold mine of useful information for biologists and drug developers, but can be challenging for non-bioinformaticians to manipulate and interpret.

The DepMap portal website (https://depmap.org/portal) provides a range of information for each gene, including (1) the cell lines and lineages dependent on the gene; (2) co-dependent genes (i.e., other genes whose effects on growth are strongly positively or negatively correlated with the gene); and (3) basal transcript abundances, copy numbers, and mutations for the gene. However, the DepMap portal has no native tools to integrate CRISPR and shRNA or to examine functional relationships among essential genes beyond pairwise comparisons.

Here we describe shinyDepMap, a web tool to enable researchers to rapidly determine the essentiality and selectivity of a given gene across cell lines and to find groups of functionally related genes with similar essentiality profiles. shinyDepMap integrates data from both CRISPR and shRNA screens, yielding robust measures of the effects of gene loss on cell viability. From these combined effect scores, we derive two measures for each gene: the degree to which loss of the gene reduces cell growth in sensitive lines ('efficacy'), and the degree to which its essentiality varies across lines ('selectivity'). To help researchers identify potential therapeutic targets, we clustered genes with strong efficacy scores into functional units, many of which represent complexes or biological pathways, as previously reported (*Pan et al., 2018*). The results of this analysis are accessible via a simple interactive web tool at https://labsyspharm.shinyapps.io/depmap.

## Results

### Assessment of consistency between CRISPR and shRNA dependency scores

The DepMap project (https://depmap.org/) provides two separate preprocessed genome-wide genetic perturbation datasets for hundreds of cell lines using either shRNA or CRISPR (*Meyers et al., 2017*; *McFarland et al., 2018*). In both datasets, the preprocessed scores represent the growth effects of knocking the gene down or out, with a strongly negative value in a particular cell line indicating essentiality. Though the preprocessing algorithms for the shRNA data take 'off-target' genes into account when generating essentiality scores, we nevertheless expected that the essentiality profiles would differ somewhat between shRNA and CRISPR due to their distinct mechanisms of reducing gene expression.

To assess the consistency between CRISPR and shRNA dependency scores, we first compared the gene/cell line combinations tested with both methods (15,847 genes in 423 cell lines, *Figure 1— source data 1*) and computed essentiality thresholds for each distribution such that a dependency score more negative than the threshold is considered essential (*Figure 1A*; 'Materials and methods'). These thresholds define the subsets of cell line/gene combinations that are determined to be essential by either CRISPR or shRNA but not both (areas A and B in *Figure 1A*). The two methods were somewhat consistent on average, with both methods yielding approximately normal dependency score distributions with mean zero and a left-skewing tail corresponding to the subset of essential genes (Pearson correlation 0.456, Spearman correlation 0.201).

Despite this concordance, the comparison highlighted differences between the two methods at the individual gene level. First, CRISPR tends to detect weak to moderate gene deletion effects more sensitively, as evidenced by the greater density of CRISPR-essential genes above the diagonal in the joint distribution plot (*Figure 1A*). For example, while both methods identify RAN, CRISPR identifies CCND1 as more essential (*Figure 1B*). Second, some genes were shown essential only by CRISPR or shRNA, but not by the other method (e.g., FOXD4 and EIF5B; *Figure 1B*).

To better understand these inconsistent dependencies, we used Fisher's exact test to determine which genes, across their perturbations in all 423 cell lines, were enriched for inconsistent dependencies. We found that 958 and 20 genes were claimed commonly essential only by CRISPR or shRNA,

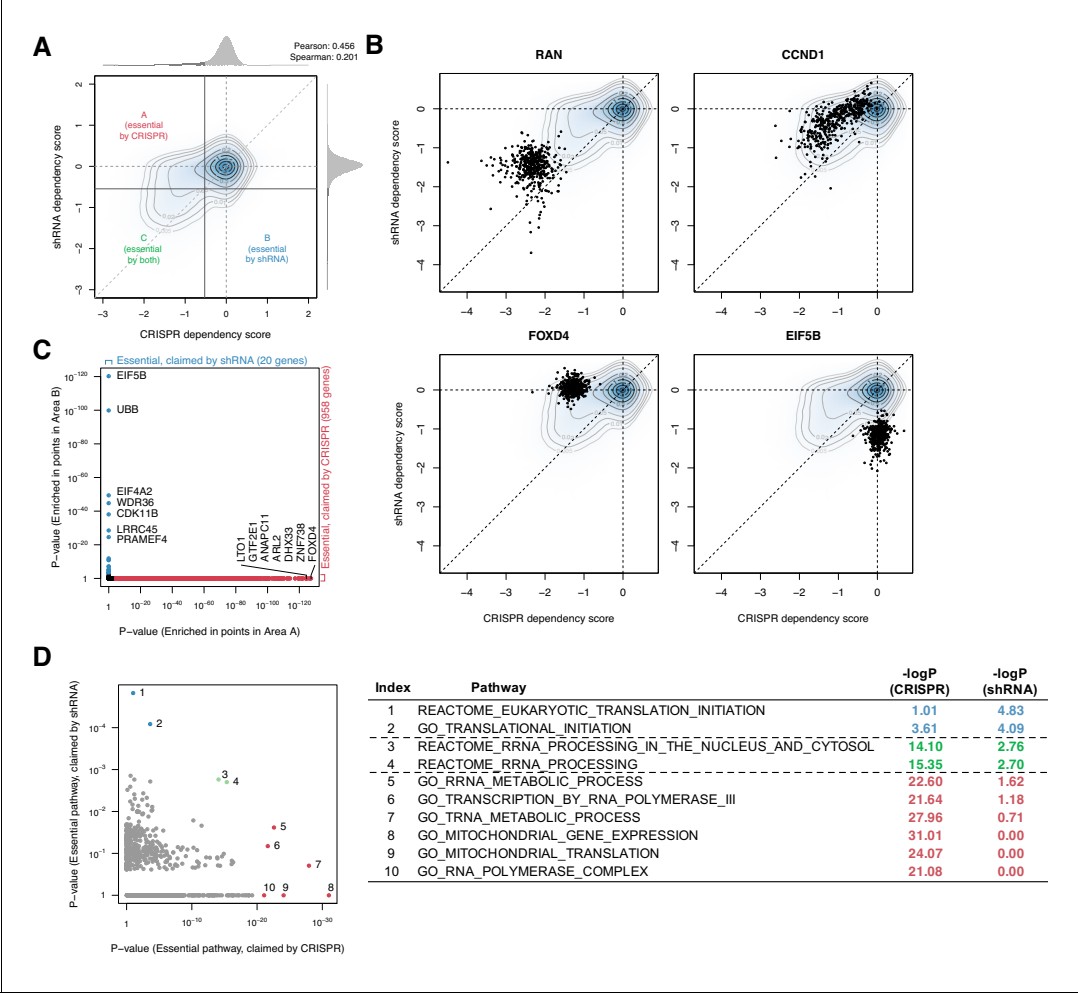

**Figure 1.** Systematic biases in CRISPR and shRNA dependency scores. (**A**) Comparison of normalized CRISPR and shRNA dependency scores of 15,847 protein-encoding genes in 423 cell lines. The density and contour plot corresponds to the distribution of the scores from all gene perturbations. Vertical and horizontal solid lines indicate the essentiality thresholds for CRISPR and shRNA dependency scores, respectively. Areas A, B, and C correspond to the regions where only CRISPR, only shRNA, or both CRISPR and shRNA claimed essential. (**B**) Comparison of CRISPR and shRNA targeting four genes. In each panel, data points correspond to each gene's perturbation in 423 cell lines. Each point corresponds to one cell line. (**C**) In total, 958 and 20 genes were claimed essential only by CRISPR or shRNA but not by the other method, respectively (Fisher's exact test, p-value<1e-3). (**D**) Assessment of the pathways overrepresented by the essential genes claimed only by CRISPR or shRNA, highlighted in C (Fisher's exact test). The online version of this article includes the following source data for figure 1:

**Source data 1.** Information of the 423 cell lines in which both CRIPSR and shRNA screening were tested.

respectively (*Figure 1C*). Notably, these two sets of inconsistently essential genes are enriched for involvement in distinct pathways: for example, tRNA metabolic process and mitochondrial translation are overrepresented in the CRISPR-only set, whereas cytosolic translation initiation is overrepresented in the shRNA-only set (*Figure 1D*). This suggests that CRISPR and shRNA have distinct biases in assessing some genes' essentiality, affecting different classes of genes. While CRISPR is considered to be less susceptible to off-target effects (*Smith et al., 2017*) and thus now generally preferred over shRNA, the results and their therapeutic relevance may also depend on the genes of interest. For example, EIF5B, a gene involved in translation initiation, is highly conserved throughout Bacteria, Archaea, and Eukarya, suggesting that it may be essential for most human cells (*Sørensen et al., 2001*). However, only shRNA, but not CRISPR, highlighted it as an essential gene. A method that can combine the two dependency scores would compensate for each other's artifacts and give more robust scores.

## A new dependency score combining CRISPR and shRNA

To summarize both CRISPR and shRNA dependency scores, $S^C$ and $S^R$, we developed a new dependency score by combining them. Recent studies have shown that similar approaches are practical (*Wang et al., 2019a*; *Gilvary et al., 2019*). Our new dependency score, $S^\theta$, was computed as the weighted average of the two such that $S^\theta = \theta S^C + (1-\theta)S^R$, where $\theta$ is the mixing ratio of the two scores. It is appropriate to have any $\theta \in [0,1]$, but we selected six values of $\theta$ :$\theta \in \{0,\ 0.2,\ 0.4,\ 0.6,\ 0.8,\ 1\}$ in this study, which are equivalent to mixing CRISPR and shRNA scores at 0:100 ($=S^R$), 20:80, 40:60, 60:40, 80:20, and 100:0 ($=S^C$). Using the equation above, we computed $S^\theta$ for 15,847 genes in 423 cell lines for each $\theta$ (*Figure 2—source data 1*). The distribution of $S^\theta$ was located between $S^C(=S^1)$ and $S^R(=S^0)$ (*Figure 2B*).

## Efficacy: gene essentiality in a sensitive cell line

We used the combined CRISPR-shRNA gene dependency scores to identify genes that are either commonly or selectively essential in the cell line panel. This distinction is important for identifying therapeutic targets because inhibition of commonly essential genes may be toxic to both cancer cells and normal cells, whereas genes that are selectively essential to particular cancers may allow for a greater therapeutic window. To capture the therapeutic potential of selectively essential genes, we characterized gene dependency effects with two parameters: the *efficacy*, which defines the strength

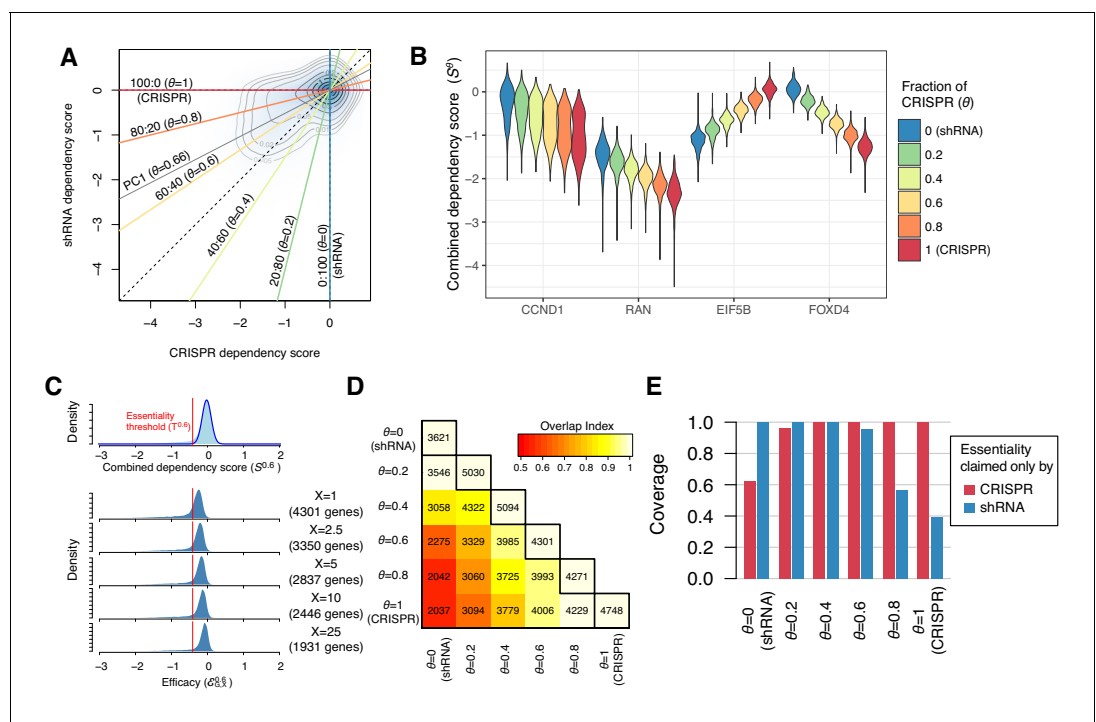

**Figure 2.** Identification of essential genes based on combined dependency score. (**A**) Dependency scores defined with different mixing ratios are computed by projecting each point onto the corresponding lines. $\theta$ denotes the fraction of CRISPR dependency scores. PC1 is the direction of the primary principal component line. (**B**) The distributions of combined dependency scores for four genes shown in *Figure 1B*. (**C**) Top panel: the distribution of the combined dependency score $S^{0.6}$. The essentiality threshold $T^{0.6}$ is determined based on this distribution. Bottom panel: the distribution of efficacy scores $\mathscr{E}_{G,X}^{0.6}$ with various X (-th percentile). Genes that satisfy $\mathscr{E}_{G,X}^{0.6} < T^{0.6}$ are defined as commonly or selectively essential, and the number of essential genes depends on X. (**D**) On the diagonal line, the numbers of commonly or selectively essential genes identified with various $\theta$ are shown. In the off-diagonal area, the numbers of essential genes identified with two distinct $\theta$ are shown. Color code indicates an overlap index, a measure of overlap between two essential gene sets. The overlap index ranges from 0 (no shared genes) to 1 (the smaller set is included in the larger set). (**E**) The extent to which the genes claimed essential by only CRISPR or shRNA are covered by the essential genes discovered by each mixing ratio. The online version of this article includes the following source data for figure 2:

**Source data 1.** The combined dependency scores for 15,847 protein-coding genes in 423 cell lines for six $\theta$.

of the effect in a sensitive cell line, and the *selectivity*, which describes the variation of the effect across cell lines.

We defined the efficacy $\mathscr{E}^\theta_{G,X}$ as the *X*-th percentile of the distribution of combined dependency scores $S^\theta$ for gene G across all cell lines, denoted $S^\theta_G$. For a given *X*, we defined a gene as essential when $\mathscr{E}^\theta_{G,X}$ is lower than the essentiality threshold $T_\theta$, which is determined from the distribution of $S^\theta$ for all genes in all cell lines (*Figure 2C*, top panel). Smaller values of *X* (lower percentiles) lead to more extreme efficacy values $\mathscr{E}^\theta_{G,X}$ and identify more essential genes in smaller subsets of cell lines (*Figure 2C*, bottom panel). We selected X=1 for most of our analysis. This is equivalent to claiming a gene essential when roughly 5 out of 423 cell lines show dependence on the gene (*p*=7.8e-5, binomial test). We discovered from 3621 to 5094 commonly and selectively essential genes from $S^\theta$ with different $\theta$ (*Figure 2D*). Reflecting the inconsistencies between CRISPR and shRNA, only 56% (2037 genes) of the essential genes overlapped between $S^C$ and $S^R$ (*Figure 2D*). As for the 958 and 20 genes claimed commonly essential only by CRISPR or shRNA (*Figure 1C*), the essential genes discovered with the same method naturally included all of them; however, the essential genes discovered with the other method included only 40–60% of them, highlighting the inconsistencies between them (*Figure 2E*). On the other hand, the combined dependency scores $S^\theta$, particularly when $\theta = 0.2,\ 0.4,\ \text{and}\ 0.6$, provide a more sensitive measure, discovering most of the essential genes claimed by either method. The first principal component line between $S^C$ and $S^R$ was parallel to the line with $\theta = 0.66$ in *Figure 2A*, which maximizes the variance of ($S^C$, $S^R$). Among the six lines, $\theta = 0.6$ (CRISPR: shRNA=60:40) is most similar to this principal component line. Therefore, we chose $\theta = 0.6$ or the corresponding $S^\theta$ primarily for the rest of the analysis and compare the performance of different $\theta$ later.

## Selectivity: the difference in gene essentiality among cell lines

We next defined selectivity, a measure of the cell line dependence of the response to the loss of a gene. Selectivity implies that gene loss has a widely varying effect across the population of cell lines, such that the dispersion of the score distribution for a selectively essential gene would be greater than that for a commonly essential gene. We defined the dispersion of gene G, $\mathscr{D}^\theta_{G,X}$, as the difference between the X-th and (100-X)-th percentiles of $S^\theta_G$, or $\mathscr{D}^\theta_{G,X} = \mathscr{E}^\theta_{G,100-X} - \mathscr{E}^\theta_{G,X}$. We found that $\mathscr{E}^\theta_{G,X}$ and $\mathscr{E}^\theta_{G,100-X}$ were related linearly for the majority of genes, corresponding to non-essential and commonly essential genes, while some genes had large positive residuals, corresponding to selectively essential genes (e.g., green vs. orange in *Figure 3A, B*). We therefore defined the selectivity $\mathscr{S}^\theta_{G,X}$ using the residuals of the (100-X)-th percentile values for $\mathscr{E}^\theta_{G,100-X}$ relative to the red regression line (*Figure 3A*), which we denote $\mathscr{R}^\theta_{G,X}$:

$$\mathscr{S}^\theta_{G,X} = \mathscr{R}^\theta_{G,X} / \widehat{\mathscr{D}^\theta_{G,X}} = \left( \mathscr{E}^\theta_{G,100-X} - \widehat{\mathscr{E}^\theta_{G,100-X}} \right) / \widehat{\mathscr{D}^\theta_{G,X}} = \left( \mathscr{D}^\theta_{G,X} - \widehat{\mathscr{D}^\theta_{G,X}} \right) / \widehat{\mathscr{D}^\theta_{G,X}}$$

where $\widehat{\mathscr{D}}^\theta_{G,X}$ is the expected dispersion of dependency scores based on the robust linear regression of $\mathscr{E}^\theta_{G,100-X}$ given $\mathscr{E}^\theta_{G,X}$, or

$$\widehat{\mathscr{D}^\theta_{G,X}} = \widehat{\mathscr{E}^\theta_{G,100-X}} - \mathscr{E}^\theta_{G,X}.$$

The expected dispersion $\widehat{\mathscr{D}^\theta_{G,X}}$ increases for more strongly negative efficacy scores $\mathscr{E}^\theta_{G,X}$, indicating greater variances in dependency effects for commonly or selectively essential genes (e.g., greater variance for RAN vs. ZCWPW1 in *Figure 3B*). This could be a result of experimental noise (e.g., fewer sequencing reads for negatively selected genes) or greater biological variability in dependency effects for these genes. To better distinguish whether genes have dispersion greater than would be expected simply based on their efficacy scores, we normalize the residual values $\mathscr{R}^\theta_{G,X}$ by dividing by the expected dispersion $\widehat{\mathscr{D}^\theta_{G,X}}$ to obtain a measure of selectivity that accounts for the variation of $\widehat{\mathscr{D}^\theta_{G,X}}$ with the X-th percentile efficacy, $\mathscr{E}^\theta_{G,X}$. Both the efficacy and selectivity of all the genes across different $\theta$ are available to download (*Figure 3C—source data 1*).

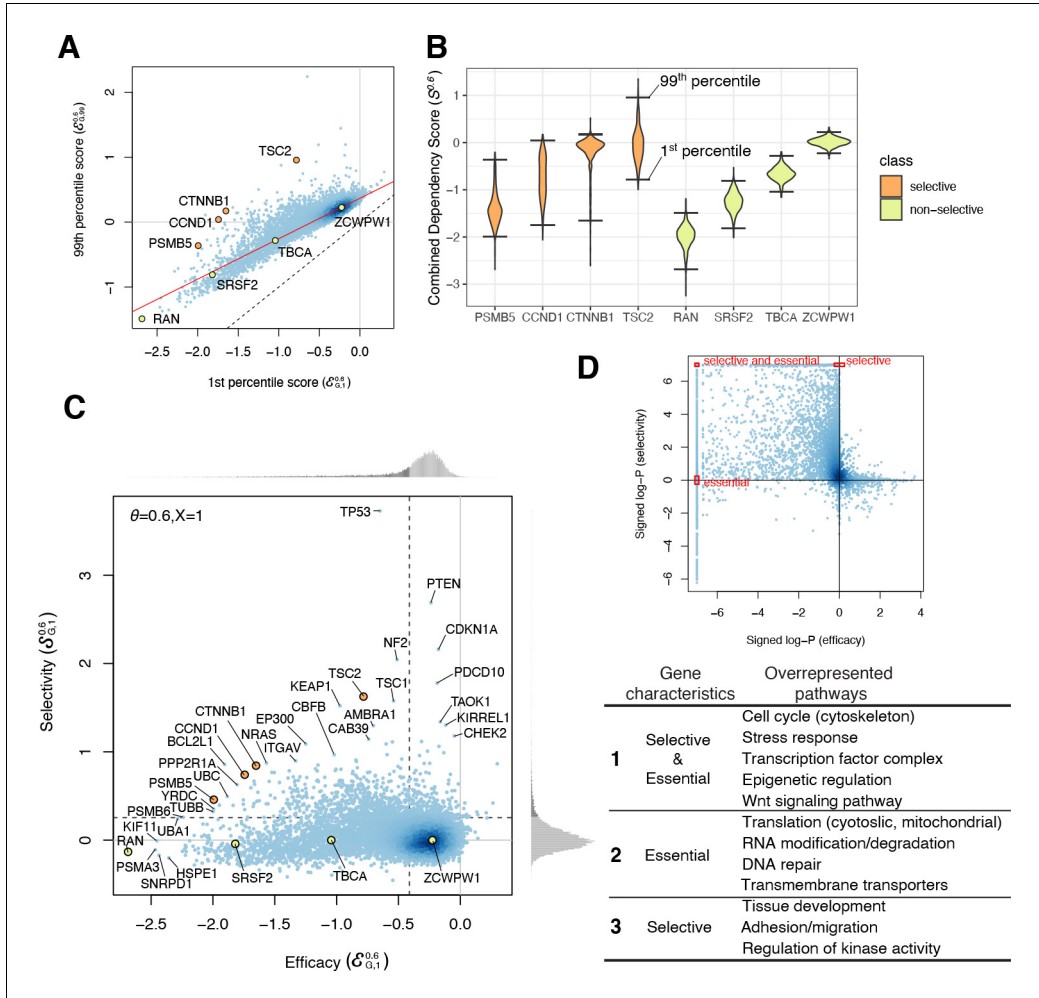

**Figure 3.** Efficacy and selectivity. (**A**) The 1st and 99th percentiles of the combined dependency score of each gene where $\theta = 0.6$. Each point corresponds to one gene. X- and Y-axes are equivalent to the efficacy with X = 1 and X = 99, respectively. Solid red line and dashed black line are robust linear regression and identity lines, respectively. (**B**) Distribution of the combined dependency scores of four selective and four non-selective genes. The 1st and 99th percentile values within each distribution are also highlighted. (**C**) The efficacy and selectivity of all genes are plotted. (**D**) Summary of the pathways overrepresented by genes with strongly negative efficacy and high selectivity, strongly negative efficacy, and high selectivity, respectively.

The online version of this article includes the following source data for figure 3:

**Source data 1.** Efficacy and selectivity for 15,847 genes for the six $\theta$ and five X: $X = \{1,\ 2.5,\ 5,\ 10,\ 25\}$.

**Source data 2.** GO/KEGG pathways overrepresented by genes with strongly negative efficacy, high selectivity, or strongly negative efficacy and high selectivity for six $\theta$ and X=1.

## Characteristics of essential genes

By comparing the efficacy and selectivity, we found that genes with strongly negative efficacy tend to be less selective, whereas selectively essential genes tend to have only moderate efficacy (*Figure 3C*). To characterize the genes that have either negative efficacy or positive selectivity, we performed gene set enrichment analysis of 6551 pathways. This revealed the pathways overrepresented among essential, selective, and both selective and essential genes. For example, chromatin regulation is overrepresented among genes that are both selective and essential; nuclear metabolism and translation are overrepresented among essential (but not selective) genes; and regulation of kinase activity and tissue development are overrepresented among selective (but not essential) genes (*Figure 3D*, *Figure 3D—source data 2*).

## Relationship of the selectivity and the lineage specificity

Cells in different lineages tend to depend on distinct essential genes compared to cells in the same lineage. We examined the extent to which lineage-specific dependence contributes to selectivity. For each gene, we assessed the relationship between the number of cell lines dependent on the gene and the gene's efficacy and selectivity, and confirmed that lower selectivity and more negative efficacy and are associated with a greater number of dependent cell lines (*Figure 4A*).

We computed the number of distinct lineages dependent on each gene using the Adaptive Daisy Model (ADaM), a permutation-based statistical model reported previously (*Behan et al., 2019*). As with dependent cell lines, a greater number of dependent lineages was associated with more negative efficacy and lower selectivity (*Figure 4B*). Overall, we found that 1050 genes are commonly essential across all the lineages, 670 genes are essential in at least one lineage, and 2581 essential genes were not lineage-dependent (*Figure 4B, C*, *Figure 4—source data 1*). Unsurprisingly, the number of dependent cell lines is strongly associated with the number of dependent lineages (*Figure 4C*).

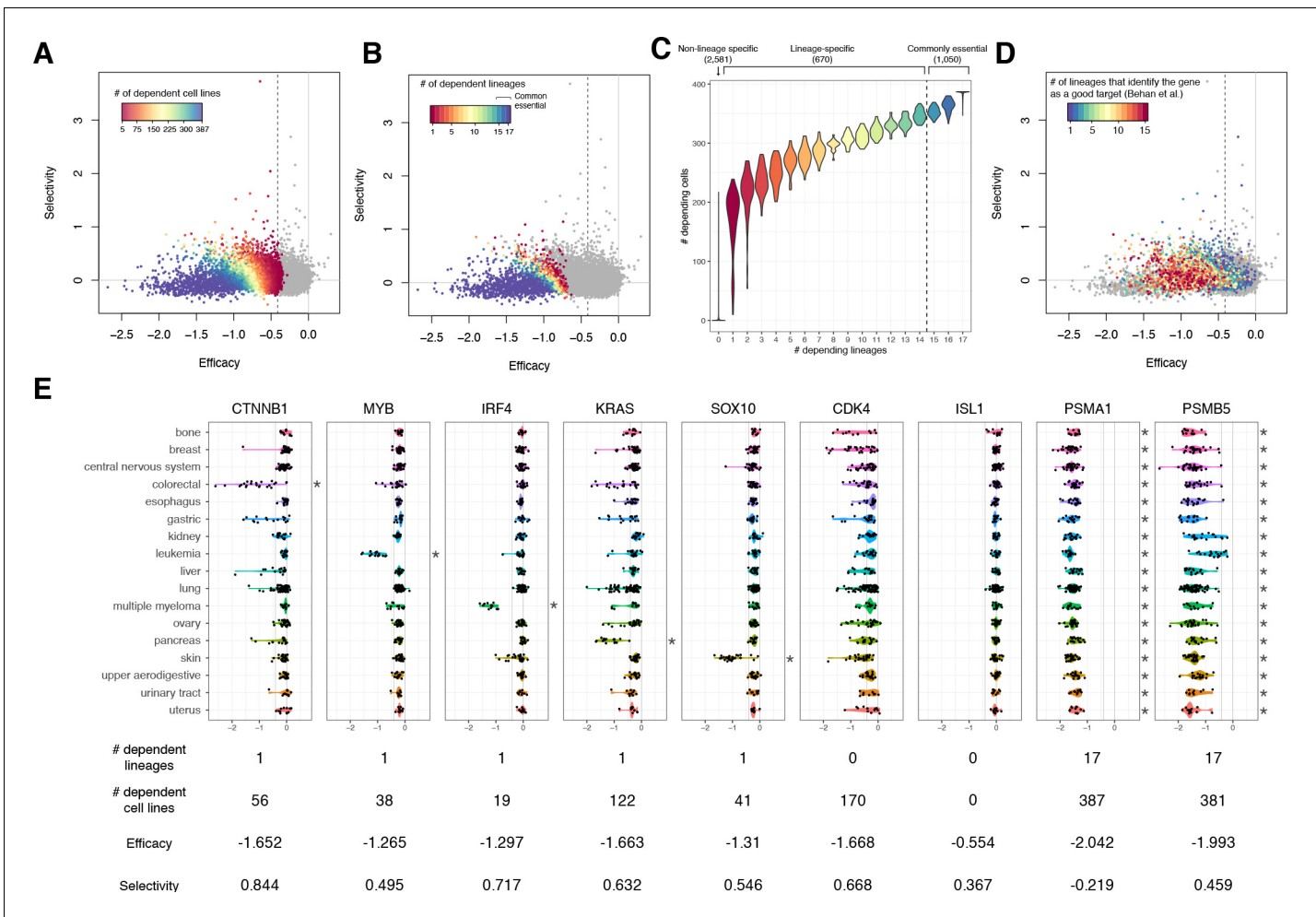

**Figure 4.** The lineage dependency. In A, B, and D, the efficacy and selectivity scatterplots (*Figure 3C*) are color-coded differently, highlighting the following properties of each gene. (A) The numbers of dependent cell lines. (B) The number of dependent lineages, computed with the Adaptive Daisy Model (ADaM). (C) The relationship between the dependent cell lines and the dependent lineages. (D) The number of lineages in which Behan et al. suggested suitable for chemotherapy targets. (E) Nine genes' dependency scores grouped by lineages, together with the number of dependent lineages and cell lines, efficacy, and selectivity. All the panels in this figure were computed using $\theta = 0.6$ and X = 1.

The online version of this article includes the following source data for figure 4:

**Source data 1.** Lineage-dependent essentiality of 17 lineages and common essentiality computed using the Adaptive Daisy Model (ADaM) for six $\theta$.

Using an independent CRISPR screening dataset, Behan et al. also proposed candidate drug targets based on selectively essential genes they identified for each lineage. We found that most genes with high selectivity scores were also identified as targets for one or more lineages in their analysis (*Figure 4D*). On the other hand, targets proposed for multiple lineages in Behan et al. (orange and red points in *Figure 4D*) tended to show moderate efficacy scores, but not necessarily high selectivity scores.

Though we saw a strong relationship between lower selectivity and a greater number of dependent cell lines and lineages (*Figure 4A, B*), we note that lineage specificity is not the only cause of high selectivity. Conventionally, high selectivity is interpreted as being commonly essential within a few lineages but non-essential in others. Some genes do manifest this type of selectivity (e.g., broad dependence on CTNNB1 in colorectal cancer, MYB in leukemia, IRF4 in multiple myeloma, KRAS in pancreatic cancer, and SOX10 in skin cancer, *Figure 4E*), but such common essentiality within a lineage is relatively unusual. In many more cases, a gene that shows high selectivity is selectively essential within each lineage (e.g., only partial dependence on CTNNB1 in liver, lung, and pancreatic cancers, or on IRF4 in skin cancer). CDK4 is a particularly strong example of this pattern as it is not commonly essential in any lineages, but is selectively essential in many. While more negative efficacy $\mathscr{E}^{\theta}_{G,X}$ is strongly correlated with larger dispersion $\mathscr{D}^{\theta}_{G,X}$ (e.g., PSMA1 vs. ISL1, *Figure 4E*; see also *Figure 3A, B*), some commonly essential genes that show similar efficacy have substantially higher selectivity than others (e.g., the selective PSMB5 vs. the non-selective PSMA1) (*Figure 4E*; see also *Figure 4A, C*). These genes that are essential in many lineages but nevertheless have high selectivity could be promising drug targets given appropriate biomarkers to characterize sensitive cell lines.

## Clustering essential genes to find related targets

The dependency profile of a gene carries information about the functions of the gene that make it essential in certain cellular contexts. When a set of genes comprise a functional unit (e.g., a pathway or a complex) that regulates cell viability, these genes would be expected to have similar dependency profiles. Gene-wise cluster analysis of the dependency data should therefore reveal functional units that connect essential genes into pathways or protein complexes. It may also be easier to relate the essentiality of pathways to cancer genotypes than to interpret the essentiality of individual genes. Pathway analysis can help identify druggable vulnerabilities at the pathway level that might be missed by single-gene analysis.

We clustered essential genes based on the similarity of the combined CRISPR-shRNA dependency scores across 423 cell lines. Our approach is based on a related pair of popular algorithms, t-distributed stochastic neighbor embedding (t-SNE), and density-based spatial clustering and noise (DBSCAN) (*van der Maaten and Hinton, 2008*; *van der Maaten, 2014*; *Ester et al., 1996*). t-SNE is a technique that reduces the dimensionality of multidimensional data while preserving the pairwise distances between data points at high dimensions as much as possible. It has been widely used for visualizing high-dimensional data, such as single-cell RNA-seq data (*Mass et al., 2016*). DBSCAN is a clustering algorithm that detects regions where the data points are gathered at high density and clusters them; it is often used to cluster data points based on their coordinates in the t-SNE plot. The combination of t-SNE and DBSCAN (expressed as 't-SNE + DBSCAN' hereafter) is a powerful clustering algorithm for high-dimensional data, such as single-cell transcriptomes (*Haber et al., 2017*).

One limitation of this approach is that the t-SNE algorithm is stochastic, producing different results and clusters with different initial seeds. However, when we compared clusters yielded by t-SNE + DBSCAN from multiple runs, we found that strongly positively correlated points are always clustered together while weakly positively correlated points are less consistently so. To obtain robust cluster assignments from t-SNE + DBSCAN, we therefore used a workflow we call *ensemble clustering* with *hierarchy over DBSCAN on t-SNE with Spearman distance matrix* (ECHODOTS).

Briefly, ECHODOTS consists of four steps (*Figure 5A*, *Figure 5—figure supplement 1*): it (1) computes the pairwise Spearman distance matrix among essential genes, (2) feeds the distance matrix as input to run t-SNE with different initial seeds 200 times, (3) clusters data points based on their coordinates in the t-SNE plot with DBSCAN, and (4) identifies the sets of genes assigned to the same cluster consistently across the 200 sets of clusters using a technique called ensemble clustering

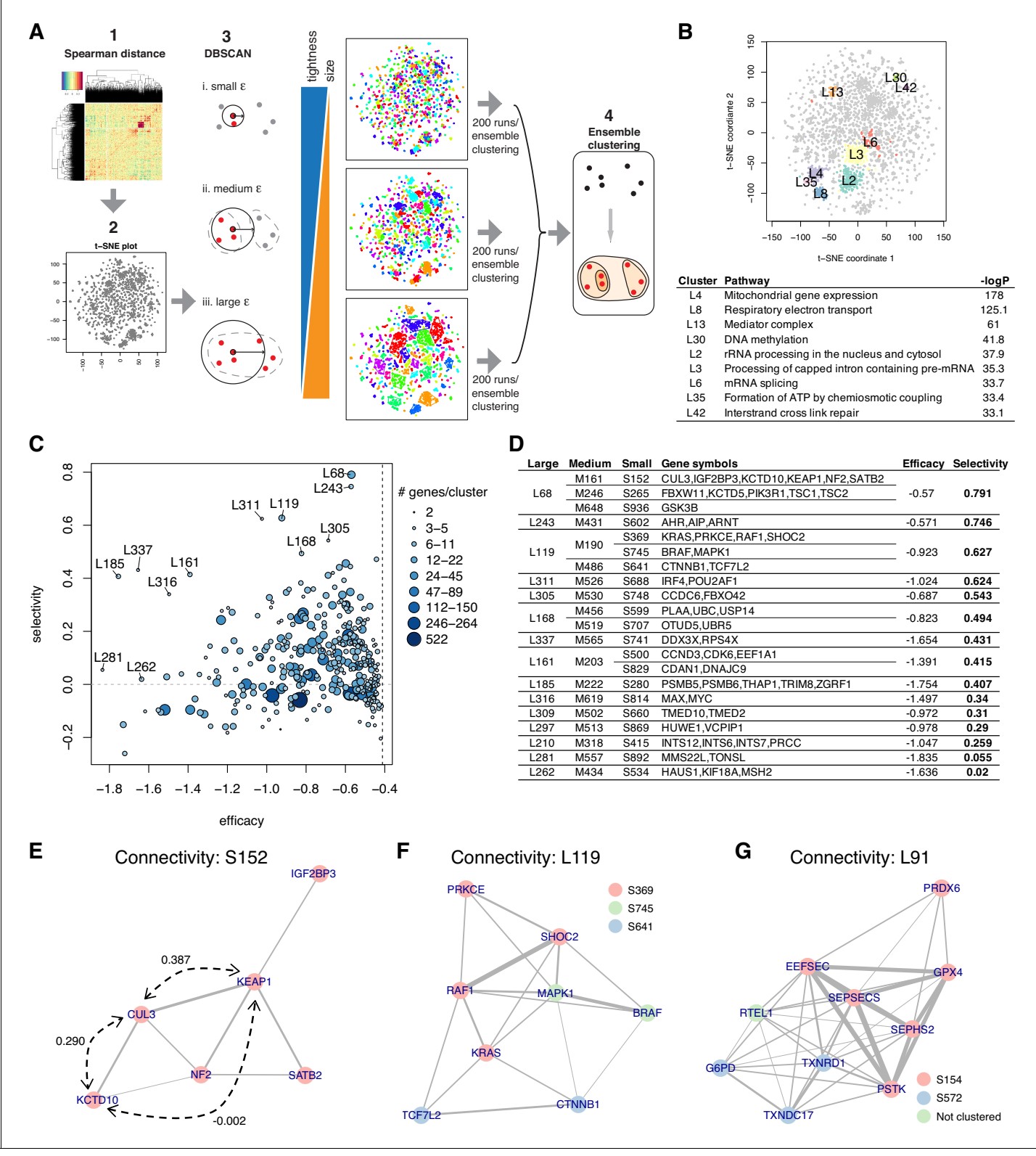

**Figure 5.** Essential gene clustering. (A) The framework of ensemble clustering with hierarchy over DBSCAN on t-SNE with Spearman distance matrix (ECHODOTS) algorithm. (B) Nine gene clusters and their associated pathways. (C) Median efficacy and selectivity of large clusters. (D) Genes consisting of large clusters with high selectivity highlighted in C. (E–G) The intra-cluster connectivity of three gene clusters as exemplars. The colors of

*Figure 5 continued on next page*

*Figure 5 continued*

nodes indicate their membership of small clusters, and the edges indicate that the two connected genes have Spearman correlation coefficient greater than 0.1. Numbers in **E** indicate Spearman correlation coefficients.

The online version of this article includes the following source data and figure supplement(s) for figure 5:

**Source data 1.** Cluster membership of essential genes and probability of their assignment to clusters for six $\theta$.
**Source data 2.** Pathways overrepresented in large clusters for six $\theta$.
**Figure supplement 1.** Ensemble clustering with hierarchy over DBSCAN on t-SNE with Spearman distance matrix (ECHODOTS) algorithm.
**Figure supplement 2.** Efficacy, selectivity, and dependent lineages with various $\theta$.
**Figure supplement 3.** Dependent cell lines and lineages using six dependency scores.

(*Hornik, 2005*). ECHODOTS produces more reliable clusters than a single run of t-SNE + DBSCAN by seeking data points that are consistently clustered together.

## Cluster reveals known and new connections among essential genes

We ran ECHODOTS against the combined dependency score $S^\theta$ of the 4301 essential genes ($\theta$=0.6, X=1) and assigned them into 879 small, 608 medium, and 338 large clusters (*Figure 5—source data 1*). Genes in the same cluster tended to be close to each other on t-SNE maps from individual runs, and some clusters were enriched for genes known to be members of specific biological pathways or complexes (*Figure 5B*, *Figure 5—source data 2*). The median efficacy and selectivity of the clusters varied widely, suggesting that some represent more promising sets of drug targets than others (*Figure 5C, D*).

In examining the clusters, we found that they often included genes that were not all mutually correlated with one another. While strongly positively correlated genes tend to be located in the same neighborhood on the t-SNE map and subsequently clustered together in ECHODOTS (*Figure 5B*), for a gene to be added to a cluster it only needs to be correlated with at least one other gene in the cluster. The structure of the correlations among the genes within a cluster can therefore highlight subtle functional relationships. For example, we plotted the correlations among six highly essential genes in cluster S152, with an edge between genes indicating a Spearman correlation greater than 0.1 (*Figure 5E*). In this cluster, KEAP1 and KCTD10 are both strongly correlated with the E3 ubiquitin ligase CUL3 (with correlations of 0.387 and 0.29, respectively), but have no correlation with each other (correlation −0.002). This is likely due to the fact that KEAP1 and KCTD10 interact with CUL3 in a mutually exclusive manner: both serve as adaptor proteins binding the same site on CUL3 but the resulting complexes degrade distinct target proteins (NFE2L2 and RHOB, respectively) (*Cullinan et al., 2004*; *Kovačević et al., 2018*).

Perhaps more intriguing are clusters that appear to show a connection between specific cellular processes and genes not otherwise known to be involved in that process. We offer two examples. One is cluster L119 (*Figure 5F*), which comprises three small clusters. Clusters S369 and S745 contain the core MAP kinase (MAPK) pathway proteins, including KRAS, RAF1, BRAF, and MAPK1, while cluster S641 consists of CTNNB1 (β-catenin) and TCF7L2, which form a bipartite transcription factor complex that is a key effector of the Wnt signaling pathway (*Jin and Liu, 2008*). These small clusters are within the same large cluster, suggesting that MAPK and Wnt signaling are functionally related in dependent cancer cell lines (*Jeong et al., 2018*). Cluster L119 also contains SHOC2, which was positively correlated to KRAS, RAF1, BRAF, and MAPK1. Multiple KRAS-mutant cancers were recently shown to be vulnerable to the loss of SHOC2 in the context of MEK inhibition, confirming the link between SHOC2 and MAP kinase pathway-driven cancers (*Sulahian et al., 2019*).

A final example is the cluster L91 (*Figure 5G*), also consisting of multiple smaller clusters. Cluster S154 contains the selenoprotein GPX4. GPX4 encodes glutathione peroxidase 4, an antioxidant enzyme, that reduces cytotoxic lipid peroxides and protects cells from a non-apoptotic cell death, called ferroptosis. Intriguingly, we found that the other genes in cluster S154 were involved in selenoprotein synthesis (SEPHS2, SEPSECS, PSTK, EFFSEC) (*Squires and Berry, 2008*), suggesting that the primary role of these genes in dependent cell lines is to synthesize GPX4. S572 contains another selenoprotein TXNRD1 and its substrate, TXNDC17 (*Espinosa and Arnér, 2019*), both of which are also strongly correlated with four selenoprotein synthesis genes. Overall, L91 seems to represent a gene set related to the sensitivity to ferroptosis (*Abdalkader et al., 2018*; *Ingold et al., 2018*).

## Comparison between different dependency scores

We have so far computed the efficacy, selectivity, and clusters of essential genes using $S^\theta$ with the fixed mixing ratio, $\theta = 0.6$, since this $S^\theta$ retains the largest variance of the original CRISPR and shRNA scores (*Figure 2A*). Here we compare the performance of different $\theta$.

For efficacy, larger $\theta$ (i.e., with a larger contribution of $S^C$ to $S^\theta$) gave more genes with strongly negative efficacy (*Figure 5—figure supplement 2A*). Consistently, the number of dependent cell lines and the number of dependent lineages per gene increased with larger $\theta$ (*Figure 5—figure supplement 2B–F*). For selectivity, larger $\theta$ gave more genes with high selectivity (*Figure 5—figure supplement 2G*). We showed earlier that essential (i.e., negative efficacy), selective (i.e., high selectivity), and both selective and essential genes overrepresent different pathways when $\theta = 0.6$ (*Figure 3D*). Similar pathways were overrepresented by selective and both selective and essential genes when $\theta = 0.8$ and 1. However, no pathways were associated with selective or both selective and essential genes when $\theta < 0.5$, suggesting that a high selectivity was given to genes more randomly (*Figure 5—figure supplement 2H*). Since it is more likely that selective genes represent certain pathways such as the ones shown in *Figure 2D*, $S^\theta$ with $\theta > 0.5$ are more reasonable ones to choose.

Next, we compared the clusters of the essential genes using ECHODOTS. Since the number of discovered essential genes varies with different $\theta$ (*Figure 2D*), we expect the number of clusters to be different. Therefore, we did not fix the number of clusters across $\theta$. Instead, we sought the upper bound of the neighborhood threshold $\varepsilon$ (termed $\varepsilon_0$) in DBSCAN for each $\theta$ because as $\varepsilon$ gets larger than a certain value, $\varepsilon_0$, most points on the t-SNE plot would start to merge to form a single large cluster (see 'Materials and methods'). We can detect incorrect merging by measuring the ratio between the first and second largest cluster sizes (*Figure 5—figure supplement 3A, B*). We found that $\varepsilon_0$ is particularly small for $\theta = 0.8$ and 1 compared to the rest of $\theta$, and more clusters were discovered for these $\theta$ consequently (*Figure 5—figure supplement 3C, D*). We compared cluster memberships of the 2008 genes identified as essential among all $\theta$ and found that $\theta > 0.5$ and $\theta < 0.5$ gave substantially different clusters (*Figure 5—figure supplement 3E*). Through the comparison, we concluded that our initial choice of $\theta = 0.6$ was a reasonable one since the combined dependency score is more informative with more weight on CRISPR than shRNA ($\theta > 0.5$, *Figure 5—figure supplement 3*) while having some contribution from shRNA is more beneficial than CRISPR alone (*Figure 2E*).

## shinyDepMap: an interactive web tool to explore the essentiality of genes

Both the clusters of essential genes and the gene efficacy and selectivity scores provide valuable information for finding potential chemotherapeutic drug targets. To make this information accessible to the broader community of experimental drug discovery researchers, we developed a web-based tool to explore these analyses, called shinyDepMap. shinyDepMap is written in R (*Chang et al., 2019*) using the shiny package for building interactive visualization tools. It consists of two apps: 'Gene essentiality' and 'Gene cluster'. Each app is a dashboard-style website (*Figure 6A*). shinyDepMap can be used in three ways: (1) via the website https://labsyspharm.shinyapps.io/depmap, (2) by downloading the code and preprocessed data from the GitHub repository (*Shimada, 2021*) and running it on a local computer, and (3) running the app from a Docker container using the image at https://hub.docker.com/r/labsyspharm/shinydepmap. The analysis workflow in the application is explained below (*Figure 6B*).

## Gene essentiality (all protein-encoding genes)

This app allows a user to explore the essentiality of all the genes tested in the DepMap genetic perturbation experiments. Its output has two panels. A scatterplot in the middle displays efficacy and selectivity scores for all genes (**3**, bold numbers correspond to the panels in *Figure 6B, C*). By hovering over the plot points with the cursor, one can find the genes corresponding to each point. When a gene name to search is provided in the input text box (**1**), corresponding genes will be highlighted in orange/red in the efficacy–selectivity plot (**3**). Genes matched with the query will be listed on the 'Matched genes' tab in the right (**4**), in which, by clicking a gene's name, the description of a gene in GeneCards (https://www.genecards.org/) will open on a new page. By further selecting a matched

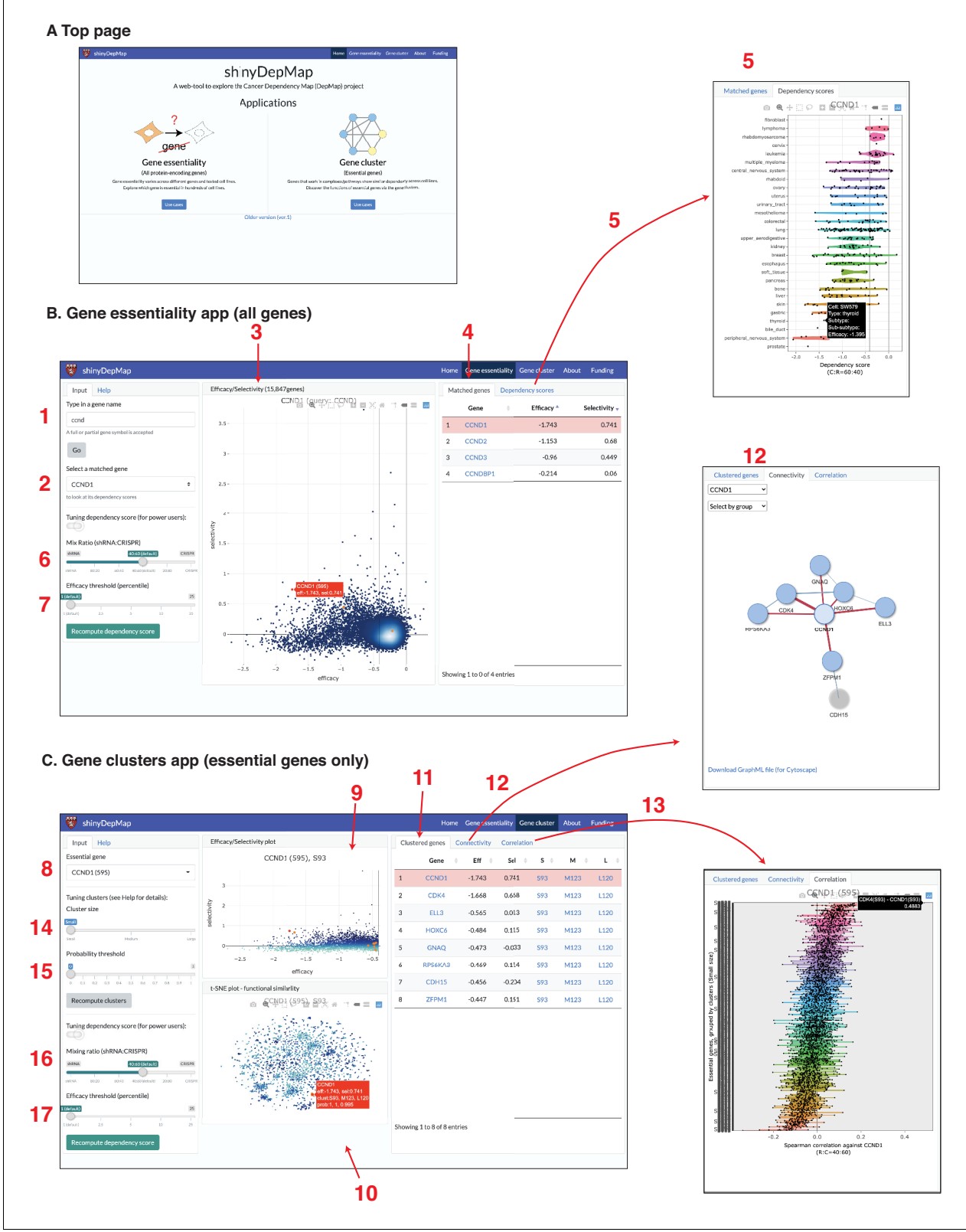

**Figure 6.** shinyDepMap: a web tool to explore DepMap dataset. (**A**) Top page. (**B**) Gene essentiality app. 1: textbox to type in a (partial) gene symbol to query; 2: dropdown menu to select a gene symbol that matches the query; 3: efficacy–selectivity scatterplot; 4: list of matched genes; 5: combined dependency score profile of the gene selected in 2; 6: mix ratio; 7: efficacy threshold (**C**) Gene clusters app; 8: dropdown menu to select an essential gene to explore; 9: efficacy–selectivity plot for essential genes; 10: t-distributed stochastic neighbor embedding (t-SNE) plot; 11: list of clustered genes;

*Figure 6 continued on next page*

*Figure 6 continued*

12: connectivity plot; 13: Spearman correlation between the selected gene and other essential genes; 14: cluster size input; 15: probability threshold input; 16: mix ratio; 17: efficacy threshold.

gene from the dropdown menu (**2**), one can see the combined dependency scores of the gene in 423 cell lines in the 'Dependency scores' tab on the right (**5**). The definition of combined dependency scores, efficacy, and selectivity can be changed by tuning the mix ratio (equivalent to $\theta$; **6**) and efficacy threshold (X-th percentile, **7**) from the input panel. We set them $\theta$ = 0.6 and X = 1 by default.

## Gene cluster (essential genes)

This app allows a user to explore gene clusters among the essential genes. When a user first selects an essential gene from the top-left dropdown menu (**8**), genes clustered with the query gene will be shown on the output panels. There are three output panels in the app. The top-center is the efficacy–selectivity plot for the essential genes (**9**). The bottom-center shows the t-SNE plot, indicating the similarity of the dependency scores among essential genes (**10**). The list of 'Clustered genes' will be shown in the right panel (**11**). 'Connectivity' tab will show how the clustered genes are connected (i.e., strongly correlated) (**12**). The graphs can be downloaded in the GraphML format. 'Correlation' tab shows the Spearman correlation coefficients between the selected gene and other essential genes grouped by the clusters (**13**). While all the genes in the small cluster are shown by default, one can change it by tuning the 'cluster size' parameter (**14**) and the probability threshold (**15**) in the left input panel. In ECHODOTS, we computed a probability at which each gene belongs to the assigned cluster. By setting the probability threshold close to one, one can show only genes that are assigned to the same clusters consistently across many runs of t-SNE + DBSCAN. This app allows the users to tune the mix ratio (**16**) and efficacy threshold (**17**) like the Gene essentiality app. Essential genes were defined based on these two parameters. Consequently, modifying these parameters affects the clusters.

## Discussion

In this paper, we described an interactive software tool, shinyDepMap, that allows users to rapidly determine the efficacy and selectivity of a gene of interest and thereby find highly selective genes that may offer promising therapeutic targets. shinyDepMap is based on both the CRISPR and shRNA genome-wide screening DepMap datasets, which we combined to generate a unified dependency score that is more informative than data from either dataset alone. Using this combined dependency score, we computed 'efficacy' and 'selectivity' scores for each gene and highlighted how these scores can be used to characterize the therapeutic potential of targeting different genes across cell lines and lineages. Finally, we performed robust clustering of commonly and selectively essential genes to highlight functional relationships. shinyDepMap allows users to interactively explore both the essentiality and clustering results and is available as a deployed web-based application at https://labsyspharm.shinyapps.io/depmap and via source code or Docker image.

Our cluster analysis of the dependency scores highlighted genes comprising complexes and pathways, as previously reported (*Pan et al., 2018*). Our research complements published work in the area of complex and pathway annotation, in part because we were able to combine both DepMap datasets. We provide cluster information in a browsable form in shinyDepMap, including the ability to tune the size of clusters. One application of this tool is 'target hopping', that is, moving from one drug target to another while keeping the selectivity profile similar (*Schenone et al., 2013*). One goal of target hopping is to identify druggable targets with similar dependency score profiles to genes of interest that are not conventionally druggable. A classic example is KRAS, which is essential to many cancers but until recently has been considered 'undruggable'. KRAS appears in cluster L119 with the druggable kinases RAF1 and MAPK1, highlighting these proteins as relevant alternative targets. One goal of shinyDepMap is to help researchers identify similar therapeutic opportunities among less well-studied genes.

Despite its potential value for therapeutic discovery, the DepMap dataset and our corresponding analysis in shinyDepMap have important practical limitations. The DepMap data characterizes the

genetic requirements for cells grown in culture, which differs from the in vivo tumor environment in critical ways: the presence of nutrient-rich media, a two-dimensional rather than a three-dimensional substrate, and the lack of a functional immune system or physiological microenvironment. In addition, the effects of genetic perturbations (knockdown and knockout) do not necessarily correspond to those of chemical inhibition, a discrepancy that makes target identification from datasets like the DepMap less straightforward (*Weiss et al., 2007*). Finally, because the DepMap includes only cultured cancer lines and no wild-type cell lines or tissues, a highly selective gene in our analysis is not guaranteed to be less toxic to normal tissues when inhibited. Provided these limitations are kept in mind, the DepMap is a powerful resource and we hope the shinyDepMap tool makes it accessible to a broad community of researchers.

## Materials and methods

### Data and code availability statement

The following data of the 2019 Q3 release were downloaded from the DepMap project website: CRISPR (avana) ('Achilles_gene_effect.csv'), combined RNAi ('D2_combined_gene_dep_scores.csv'), and the cell line metadata ('sample_info.csv'). The CRISPR and shRNA efficacy data provided by the DepMap project were normalized with CERES and DEMETER2 algorithms by the Broad Institute, respectively. To compute the combined dependency score, we use the data of 15,847 genes in 423 cell lines, which were examined with both CRISPR and shRNA. The codes generated during this study are available at https://github.com/kenichi-shimada/depmap-analysis (data processing and analysis) and https://github.com/kenichi-shimada/shinyDepMap (standalone shinyDepMap). This study did not generate unique datasets.

### Imputing missing values in shRNA and CRISPR dependency scores

We first compared all the conditions (genes and cell lines) that were using both CRISPR and shRNA in a scatterplot. In total, 15,847 genes were perturbed using both methods in 423 cell lines, and we compared 4,846,055 conditions that were non-missing values. Here, one condition is defined as a perturbation of one gene, either using CRISPR or shRNA in one cell line. We next imputed missing values in CRISPR and shRNA datasets using non-missing data from the other method. In total, 1,345,642 conditions (20%) were tested using CRISPR, but not shRNA. Also, 19,001 (0.28%) were tested using shRNA, but not CRISPR. In these cases, the missing values were imputed from the other data using local polynomial regression (loess) function in R. In total, 4457 (0.066%) conditions were not tested by either CRISPR or shRNA, which were left as missing. After missing values were imputed, the combined dependency score from two values was computed.

### Computing the dependency score combining CRISPR and shRNA data

In this method section, we use $S_{G,L}^{\theta}$ instead of $S^{\theta}$ to represent the dependency score of gene *G* in cell line *L* to make the argument clearer. To summarize CRISPR and shRNA dependency scores, we developed a new dependency score of a gene *G* in a cell line *L*, $S_{G,L}^{\theta}$, using the following equation:

$$S_{G,L}^{\theta} = \theta S_{G,L}^{C} + (1-\theta)S_{G,L}^{R} \quad (0 \leq \theta \leq 1)$$

where $S_{G,L}^{C}$ and $S_{G,L}^{R}$ are the dependency scores of the gene *g* in a cell line *l*, given by CRISPR and shRNA alone, respectively. The combined dependency score is a function of the mixing value $\theta$. We particularly chose and compared six values of $\theta$ : $\theta \in \{0, 0.2, 0.4, 0.6, 0.8, 1\}$. The resulting combined dependency scores, $S_{G,L}^{\theta}$, were computed for 15,847 genes in 423 cell lines for each $\theta$.

### Defining essential conditions

We defined the loss of a gene *G* is essential in a cell line *L* when the score $S_{G,L}^{\theta}$ is lower than the essentiality threshold $T_{\theta}$: $S_{G,L}^{\theta} < T_{\theta}$. $T_{\theta}$ is defined as follows: we first fit the kernel density estimate function to the entire distribution of the dependency scores across all genes and cell lines, or $S_{\cdot,\cdot}^{\theta}$. This distribution is well fitted with a normal distribution with a heavy left tail. We computed the

mean $\mu$ and standard deviation $\sigma$ from the right-half of the data and defined $T_\theta$ such that $P\left(S_{G,L}^\theta < T_\theta\right) = 0.001$, where $S_{G,L}^\theta \sim N(\mu, \sigma)$.

## Identifying inconsistent essential genes between CRISPR and shRNA

To find out the genes that are claimed essential only by CRISPR or shRNA, we first computed $T_\theta$ for CRISPR and shRNA, or $T_C$ and $T_R$. Next, we sought the set of scores targeting the same gene in all the cells ($S_{G,\cdot}^C$, $S_{G,\cdot}^R$). The essentiality claimed only by CRISPR or shRNA was expressed as $S_{G,\cdot}^C < T_C \bigcap S_{G,\cdot}^R \geq T_R$ and $S_{G,\cdot}^C \geq T_C \bigcap S_{G,\cdot}^R < T_R$, respectively. They are illustrated as areas A and B in *Figure 1B*. Using one-tailed Fisher's exact test, we computed the statistical significance of the enrichment of the data points ($S_{G,\cdot}^C$, $S_{G,\cdot}^R$) in the areas A and B and found that 958 and 20 genes were claimed essential only by CRISPR and shRNA, respectively. We also computed the statistical significance of the overlap between these gene sets with publicly available gene annotations, Molecular Signature Database (MSigDB v7.0) (*Subramanian et al., 2005*), using Fisher's exact test.

## Computing the efficacy and selectivity for each gene

The efficacy $\mathscr{E}_{G,X}^\theta$ measures how essential a gene is in a sensitive cell line:

$$\mathscr{E}_{G,X}^\theta = \text{the X}^{\text{th}} \text{ percentile of } S_{G,\cdot}^\theta \qquad (X \in \{1, 2.5, 5, 10, 25\})$$

We defined a gene as essential for a given X when $\mathscr{E}_{G,X}^\theta < T_\theta$ (*Figure 3A*).

To define the selectivity, we first determined the dispersion of the distribution of $S_{G,\cdot}^C$, $\mathscr{D}_{G,X}^\theta$:

$$\mathscr{D}_{G,X}^\theta = \mathscr{E}_{G,100-X}^\theta - \mathscr{E}_{G,X}^\theta$$

$\mathscr{E}_{G,X}^\theta$ and $\mathscr{E}_{G,100-X}^\theta$ were in a strong linear relationship for most of the genes, which correspond to commonly essential genes, while some genes have large positive residuals. We defined the residual from the regression line, $\mathscr{R}_{G,X}^\theta$, as follows:

$$\mathscr{R}_{G,X}^\theta = \mathscr{D}_{G,X}^\theta - \widehat{\mathscr{D}_{G,X}^\theta}$$

$$\widehat{\mathscr{D}_{G,X}^\theta} = \widehat{\mathscr{E}_{G,100-X}^\theta} - \mathscr{E}_{G,X}^\theta = f\left(\mathscr{E}_{G,X}^\theta\right) - \mathscr{E}_{G,X}^\theta$$

and the selectivity of the gene G, $\mathscr{S}_{G,X}^\theta$, was defined as

$$\mathscr{S}_{G,X}^\theta = \mathscr{R}_{G,X}^\theta / \widehat{\mathscr{D}_{G,X}^\theta}.$$

## Overlap of essential genes among different $\theta$

The number of essential genes, defined as $\mathscr{E}_{G,1}^\theta < T_\theta$, depends on the mixing ratio $\theta$. To assess the overlap between the essential gene sets across different $\theta$, we computed an overlap index for any pairs of $\theta$ that is denoted as $O_X(\theta_1, \theta_2)$:

$$O_X(\theta_1, \theta_2) = N_X^{\theta_1 \cap \theta_2} / \min\left(N_X^{\theta_1}, N_X^{\theta_2}\right)$$

where $\theta_1$ and $\theta_2$ are specific values of $\theta$, $N_X^{\theta_1}$ is the number of essential genes when $\theta = \theta_1$, and $N_X^{\theta_1 \cap \theta_2}$ is the number of shared genes between the two essential gene sets. By definition, $O_X(\theta_1, \theta_2)$ can take any values between 0 and 1: the index is zero when the two essential gene sets do not share any genes; the index is one when the two essential genes are identical (*Figure 2D*).

## Identifying pathways overrepresented by essential and/or selective genes

For each of essential (i.e., negative efficacy), selective (i.e., high selectivity), or both selective and essential genes, we sought pathways that were overrepresented by each gene. We sorted all the genes by efficacy and selectivity in descending order and ran gene set enrichment analysis (GSEA)

with the sorted genes against the pathways from MSigDB. GSEA was performed utilizing fgsea package with $10^7$ permutations (*Sergushichev, 2016*).

## Identifying lineage-specific and universally essential genes

To compute the lineage specificity, we utilized ADaM. ADaM calculates the minimum number of dependent cell lines that are required for a gene to be considered as commonly essential among the cell lines in question (*Behan et al., 2019*). It is implemented in the ADAM2 R package (https://github.com/DepMap-Analytics/ADAM2). The dependency score matrix, $S_{\cdot,\cdot}^{\theta}$, contains the information of 423 cell lines representing 28 lineages. We focused on a subset of 387 cell lines in 17 lineages that includes 10 or more cell lines. We computed the binary essentiality matrix for each G and L, where 1 if $S_{G,L}^{\theta} < T_{\theta}$ and 0 otherwise. We then provided a subset of the matrix attributed to each lineage as input and calculated the minimum number of dependent cell lines for the lineage. Each lineage is considered dependent on *G* when the number of dependent cell lines in the lineage is equal to or greater than the minimum number of dependent cell lines. To compute universally essential genes utilizing ADaM, we calculated the binary essential matrix for the 17 lineages instead of cell lines and the minimum number of dependent lineages providing the matrix as input.

Behan et al. also provide a list of genes that are good targets for chemotherapies for each lineage. We counted the number of lineages they suggested was a good target for each gene and mapped them onto the efficacy/selectivity plot (*Figure 4D*).

## Robust cluster analysis utilizing t-SNE and DBSCAN: ECHODOTS

We implemented a new cluster algorithm, ECHODOTS, extending the combination of t-SNE and DBSCAN. It is graphically summarized in *Figure 5A*, and its pseudocode is provided in *Figure 5—figure supplement 1*.

First, we computed the Spearman distance matrix or 1-Spearman correlation coefficient across all pairs of essential genes (line (1) in *Figure 5—figure supplement 1*). This matrix was provided as input, and the coordinates of each data point in a 2D plane were computed with t-SNE (line (2)). Next, we clustered data points based on their coordinates with DBSCAN such that any two points whose distance is smaller than the neighborhood threshold ε are assigned into the same cluster (line (4)). We note that the range of the coordinates, *L*, varies among different runs of t-SNE. It is more reasonable to make the denominator *d* constant rather than to make ε constant; therefore, we determined ε relative to *L*: $\varepsilon = L/d$ (line (3)). The resulting cluster set *C* was computed by DBSCAN using $\varepsilon$ derived from the preset parameter *d* (line (4)), which we set an integer ranging from 30 to 200. We ran t-SNE and DBSCAN to compute *C* 200 times using different initial seeds.

Next, we computed consistent clusters *CC* across 200 cluster sets *C* for each *d* (or equivalently ε) using ensemble clustering available in clue R package (line (5)). When *d* is too small, (ε is too large), most of the points are erroneously connected. The mean size and total number of consistent clusters depend on the neighborhood threshold ε. The smallest ε yields the largest number of small clusters, in which data points within each cluster are most tightly connected. The largest ε yields the smallest number of large clusters, in which data points within each cluster are most loosely connected. When ε is too large (correspondingly, *d* was too small), most genes form one massive cluster in an extreme case. To avoid this, we set a lower bound for *d* or equivalently upper bound for ε such that $\varepsilon = L/d \leq L/d_0 = \varepsilon_0$ (line (6)). We determined $d_0$ by looking at the ratio between the sizes of the first and second largest clusters (*Figure 5—figure supplement 3A*). When ε gets smaller (*d* gets larger), clusters get smaller and tighter. With smaller ε, some genes are not clustered with any other genes. We called these genes forming clusters only by themselves 'noise' and distinguished them from other clusters containing more than one gene. Eventually, all the genes become isolated or noise, but we stopped our analysis far ahead (*d* = 200). When $\theta = 0.6$, we chose three different values of *d* (*d* = 65, 100, 141), which discovered 338, 608, and 879 clusters (line (7)). Most of the smaller clusters derived with larger *d* were contained in the larger clusters derived with smaller *d*. Therefore, we constructed a hierarchical relationship between the three sets of clusters. There were a few cases where a gene belongs to distinct clusters with different *d*, but the smaller cluster is not a part of the larger cluster and thus the hierarchy is not constructed.

Note that DBSCAN is a hard clustering algorithm that assigns each gene into only one cluster. Once clustered, both strongly and weakly correlated genes become indistinguishable in hard

clusters. On the other hand, ECHODOTS performs a soft clustering that can assign each gene to more than one cluster. It conducts the 'majority vote' among 200 runs of the clustering and computes the probability of a point being assigned to each cluster, which provides us with the strength of evidence for the cluster assignment of each gene. Thus, ECHODOTS produces more reliable clusters than a single run of t-SNE + DBSCAN by seeking data points that are consistently clustered together.

## Implementation of shinyDepMap website

shinyDepMap was built using shiny package. The following packages are also used to implement the tool: ggplot2 (v3.2.1), RColorBrewer (v1.1–2), shinyWidgets (v0.4.8), plotly (v4.9.0), DT (v0.8), visNetwork (v2.0.8), tibble (v2.1.3), dplyr (v0.8.3), and tidyr (v0.8.3). shinyDepMap can be run locally without an internet connection. One can download the code and data at https://github.com/kenichi-shimada/shinyDepMap, and run locally following the link's instruction.

## Acknowledgements

The authors thank Laura Maliszewski, Peter Sorger, and Rebecca Ward of Harvard Medical School for helpful discussions. This work was financially supported by the Japan Society for the Promotion of Science Overseas Research Fellowship (to KS), NIH R35GM131753 (to TJM), and P50GM107618.

# Additional information

## Competing interests

John A Bachman: has received consulting fees from Two Six Labs, LLC. The other authors declare that no competing interests exist.

## Funding

| Funder | Grant reference number | Author |
| --- | --- | --- |
| Japan Society for the Promotion of Science | H29-814 | Kenichi Shimada |
| National Institutes of Health | P50GM107618 | Kenichi Shimada Timothy J Mitchison |
| National Institute of General Medical Sciences | R35GM131753 | Timothy J Mitchison |
| Defense Advanced Research Projects Agency | W911NF-15-1-0544 | John A Bachman |
| National Cancer Institute | U54-CA225088 | Jeremy L Muhlich |

The funders had no role in study design, data collection and interpretation, or the decision to submit the work for publication.

## Author contributions

Kenichi Shimada, Conceptualization, Data curation, Software, Formal analysis, Funding acquisition, Investigation, Visualization, Methodology, Writing - original draft, Project administration, Writing - review and editing; John A Bachman, Writing - review and editing; Jeremy L Muhlich, Software; Timothy J Mitchison, Conceptualization, Supervision, Writing - review and editing

## Author ORCIDs

Kenichi Shimada (iD) https://orcid.org/0000-0001-8540-9785
John A Bachman (iD) https://orcid.org/0000-0001-6095-2466
Jeremy L Muhlich (iD) http://orcid.org/0000-0002-0811-637X
Timothy J Mitchison (iD) http://orcid.org/0000-0001-7781-1897

Decision letter and Author response
Decision letter https://doi.org/10.7554/eLife.57116.sa1
Author response https://doi.org/10.7554/eLife.57116.sa2

## Additional files

### Supplementary files
• Transparent reporting form

### Data availability

Data files have been provided for Figures 1, 3, 4, and 5 on FigShare: https://figshare.com/projects/shinyDepMap_Source_Data/97382 (DOIs: https://doi.org/10.6084/m9.figshare.13653251.v1, https://doi.org/10.6084/m9.figshare.13653257.v1, https://doi.org/10.6084/m9.figshare.13653260.v1, https://doi.org/10.6084/m9.figshare.13653266.v1, https://doi.org/10.6084/m9.figshare.13653272.v1, https://doi.org/10.6084/m9.figshare.13653278.v1, https://doi.org/10.6084/m9.figshare.13653281.v2).

The following datasets were generated:

| Author(s) | Year | Dataset title | Dataset URL | Database and Identifier |
|---|---|---|---|---|
| Shimada K | 2021 | shinyDepMap - Source Data 1 | https://doi.org/10.6084/m9.figshare.13653251.v1 | figshare, 10.6084/m9.figshare.13653251.v1 |
| Shimada K | 2021 | shinyDepMap - Source Data 2 | https://doi.org/10.6084/m9.figshare.13653257.v1 | figshare, 10.6084/m9.figshare.13653257.v1 |
| Shimada K | 2021 | shinyDepMap - Source Data 3 | https://doi.org/10.6084/m9.figshare.13653260.v1 | figshare, 10.6084/m9.figshare.13653260.v1 |
| Shimada K | 2021 | shinyDepMap - Source Data 4 | https://doi.org/10.6084/m9.figshare.13653266.v1 | figshare, 10.6084/m9.figshare.13653266.v1 |
| Shimada K | 2021 | shinyDepMap - Source Data 5 | https://doi.org/10.6084/m9.figshare.13653272.v1 | figshare, 10.6084/m9.figshare.13653272.v1 |
| Shimada K | 2021 | shinyDepMap - Source Data 6 | https://doi.org/10.6084/m9.figshare.13653278.v1 | figshare, 10.6084/m9.figshare.13653278.v1 |
| Shimada K | 2021 | shinyDepMap - Source Data 7 | https://doi.org/10.6084/m9.figshare.13653281.v2 | figshare, 10.6084/m9.figshare.13653281.v2 |

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
