## [Decision Letter]

**Acceptance summary:**

The main strength of the work is combining into a single measure CRISPR and shRNA screening data from DepMap, and providing a very convenient interface to explore these data. The approach is well-considered, and the trade-off between streamlining the exploration vs increased difficulty in interpretation is reasonable. The web-based interface is easy to use, and most of the limitations in the data analysis are clearly described. As such, the tools could be used by the community investigating gene function.

**Decision letter after peer review:**

Thank you for sending your article entitled "shinyDepMap: A tool for browsing the Cancer Dependency Map reveals functional connections between genes" for peer review at *eLife*. Your article is being evaluated by three peer reviewers, and the evaluation is being overseen by a Reviewing Editor and Maureen Murphy as the Senior Editor.

As you can see from the below comments, at least two of the reviewers identify potential significance in your study. However, all of the reviewers have identified multiple weaknesses in the submitted work, bearing on the capacity of the tool to yield a measurable benefit over what is currently available; the significance of the biological insights reported; and the clarity of the presentation.

Reviewer #1:

The main strength of the work is combining into a single measure CRISPR and shRNA screening data from DepMap and providing a very convenient interface to explore these data. The approach is plausible and the trade-off between streamlining the exploration vs increased difficulty in interpretation is reasonable. The web-based interface is easy to use, and most of the limitations in the data analysis are clearly described. As such, the tools could be used by the community and the paper deserves to be published after a revision.

The biggest problem with the paper itself is the way it is written, which appears to be in a big haste. Multiple terms are used to describe the same thing (like "dependency score" and "perturbation score"). Sometimes the sentences appear to have self-contradictions (like "The median cluster size is 6, 4, and 3 genes for small, medium, or large clusters, respectively"). References in text to figure panels are frequently not matching (e.g., the sentence "We developed a new gene dependency metric, termed "perturbation score" " refers to Figure 2B, which is just a set of violin plots showing "The perturbation score profiles of the same four genes as in A."). Words missing in sentences make them hard to understand e.g., in the sentence "such that the size of the 1st and 2nd largest clusters {less than or equal to} b (b = 2.26 in our case " the word "ratio" is missing). Grammar and word usage issues additionally hamper difficulty in understanding.

Perhaps most importantly, the document is really poorly structured with the definitions given after the analysis using the defined term was described, some of relevant logic found in Materials and methods section etc. While having the details of informatics analysis would be rather illuminating for the "average biology scientist", the paper still should be rewritten and streamlined to provide the better flow and eliminate the elements of sloppiness, including all points mentioned above across the main text, figures and figure legends.

1) Authors' definition "We defined non- essential genes such that they are not identified as essential from the earlier analysis using the perturbation scores of the 10 , 20 and 42 most sensitive cell lines, as well as not identified as promoting its growth using the scores of the 10 , 20 and 42 least sensitive cell lines." should be justified – why was it important to exclude also genes which appear to promote growth upon knockdown from the set of non – essential genes? How many of these genes were excluded?

2) DepMap genes may have designations as "Strongly Selective" and/or "Common Essential". What is the relation between genes identified as such in this study and in DepMap? A supplementary figure (Venn diagrams) and/or table may illustrate the overlap.

3) It is not quite clear what in Figure S1B is illustrated which would be not clear from S1A.

4) Selectivity is already a very vague notion, because it can arise from differences between cell types, or from the presence/absence of mutation in certain gene(s), etc. Median selectivity, used in Figure 4F, is trickier still, because different genes in the cluster may have different sources of selectivity. While Figure 4F is, indeed, a good way to illustrate a diversity of identified clusters, a word of caution would help to avoid overinterpreting the results.

5) In the application itself, Gene Clusters are very nicely presented in the Connectivity view. It would be very helpful to provide an option to save the underlying table with gene names and correlations (for example, to enable the subsequent use in Cytoscape). At the bare minimum, some way of exporting at least gene names in the cluster should be provided, apart from copying the whole table from "Clustered genes" view and then removing unused columns.

6) CRISPR data seems to dominate the perturbation score. How much different the clustering based on the perturbation score is if compared to using CRISPR data alone? Could any independent measure be provided to compare the sets of clusters? E.g., distribution of connectivity scores based on protein-protein interactions, or of enrichment scores?

7) In the application Gene Clusters , Correlation View, the names of the clusters can be discerned when the user zooms in. However, in the static view (Figure 1C, section 12) it should be indicated in the legend that the three vertical "lines" on the left are, in fact, the cluster labels.

Reviewer #2:

Shimada at al. present a combination CRISPR and shRNA screening data from DepMap and built user-friendly browser (https://labsyspharm.shinyapps.io/depmap) for the exploration of the dataset using their metric of efficacy and selectivity.

The efficacy is the "perturbation score" of the 10th most sensitive cell line for each gene, whereas selectivity is an ad-hoc measure that the authors define according to the dispersion of the efficacy between most sensitive and least sensitive cell lines. The app also allow to explore a clustering of the data computer with their algorithm ECHODOTS.

The DeepMap dataset is a very important resource to identify candidate targets and cancer dependency and efficient methods that allow exploration and analysis of these data are welcome. Unfortunately the paper of Shimada does not reach neither of these aims.

Overall the paper is very poorly written and very difficult to follow. It starts with the presentation of the shiny app for the 2,492 essential genes that they select with their method. I understand that the aim is to allow non-programmers to access the data, the problem is the kind of analysis supporting the plots that, I think, should much more improved.

The author claim to compute a novel method of the efficacy, a new methods to select genes which are selectively lethal for cell line, and also a novel clustering method. Unfortunately none of these methods is sufficiently justified by the authors of compared with the state of the art of at least the current literature. For example, they could have compared their selectivity method to the ADaM method reported in (Behan et al., 2019), moreover it is not clear why the efficacy score computed on the basis of the "perturbation score" is more robust of should be preferred with respect to other approach. Finding out how they defined the efficacy was a very difficult.

Regarding the novel clustering algorithm that they propose, it is very strange that the authors need to develop a novel clustering algorithms given the plethora of available methods in the literature. Why should they develop a new one and what limits of the current state of the art their algorithm should overcome it completely obscure. Moreover, from their description of the method I doubt it is really a clustering procedure at all.

There is no guarantee that rule they describe "It assigns two genes into the same cluster when the distance between the genes is smaller than epsilon" you have that each point belongs to just one cluster. The authors do not specify to what data the clustering is applied. CRISPR? shRNA? Both?

Moreover, the choice of the parameters is rather arbitrary and not discussed in the paper.

Overall, I suggest the authors to focus on what they think could be an improvement of the state of the art in at least one on the three areas (efficacy selectivity and clustering) and convince the reader that on what and why they are improving something.

Regarding the shiny app itself, it is a very basic exploratory tool. A very interesting improvement that could be useful for target identification and exploitation of synthetic lethality of to include genomic information to check if the cells depending on a specific gene share some common molecular features (mutations, copy number, pathways). This could for example be done using the efficacy as an independent variable and the molecular features as predictors in the cells depending on a specific gene.

Reviewer #3:

In this manuscript, Shimada et al. develop a tool called shinyDepMap to enable the non-expert bioinformatician to analyze cancer dependency datasets through a user friendly interface. This tool reports gene perturbation scores, which combines essentiality scores from shRNA and CRISPR based screens for various cell lines and utilizes network analysis to identify gene cluster and essential pathways in the various cancer cell types. Overall, this is a very useful tool to the community at large, especially those with little computational background – similar to cbioportal. It will likely be widely adopted however I have a few suggestions for improvement:

Overall the website is slow. I assume this is a test version, however if it is to be widely used, some effort will be needed to ensure it is able to meet the large bandwidth and demand. Ideally it should also be hosted in an independent weblink (e.g. cbioportal).

It would be helpful if in addition to the interactive graphs which are very helpful, if the tool would export data in PDF/JPG for publication purposes as well as a tabular format for each cell line/perturbation score. It is sometimes difficult to identify all key genes/cell lines simply by hovering over the graphs and it is helpful to be able to obtain the output in a table format.

It would be helpful if the perturbation scores can include a separate parameter to indicate whether the CRISPR and shRNA based essentiality scores significantly diverge. The authors explore this in some depth and report that their score is at least as good as the CRISPR screen, however it would be helpful to the user if they received some form of confidence score that indicates how well the two modalities agree/disagree. There are many reasons for discrepancy and it would be helpful to be aware of such a discrepancy when it exists.

While the authors perform robustness analysis when choosing their cutoff to define essential genes (10th most sensitive cell line) it is unclear still why they picked such a cutoff. Perhaps some additional explanation of the rationale would be helpful. Also in cancer types where there are more cell lines available would it be helpful to define the 10th percentile rather than the 10th cell line?

I find the selectivity analysis summarized in Figure 3E to be perhaps the most interesting biological insight from this paper. This is important therapeutically. However the way it is phrased, selectivity compares perhaps most likely different tumor types (i.e. a drug that's toxic to breast cancer cells is less toxic to sarcoma cells for example). Therapeutically this comparison is less valuable than comparing normal to transformed/cancer cell line for each histology or on a pan cancer level. If the data exists, it would be extremely helpful to recompute selectivity comparing cancer/non-cancer cells and reproducing Figure 3E for such a comparison.

Finally, it would be very helpful for the user to be able to select a subset of cell lines to perform pairwise analysis. For example if a user would like to ask what genes are essential for cell lines harboring a KRAS mutation vs cells without a KRAS mutation, having the ability to perform pairwise analysis and displaying the data for pre-selected cell lines based on a specific feature/histology would be extremely helpful and significantly expand the value of this tool. I realize that this would necessitate incorporating the mutational annotation and copy number annotation of each cell line but perhaps the effort would be well worth it if feasible.

---

## [Author Response]

Reviewer #1:The main strength of the work is combining into a single measure CRISPR and shRNA screening data from DepMap and providing a very convenient interface to explore these data. The approach is plausible and the trade-off between streamlining the exploration vs increased difficulty in interpretation is reasonable. The web-based interface is easy to use, and most of the limitations in the data analysis are clearly described. As such, the tools could be used by the community and the paper deserves to be published after a revision.The biggest problem with the paper itself is the way it is written, which appears to be in a big haste. Multiple terms are used to describe the same thing (like "dependency score" and "perturbation score"). Sometimes the sentences appear to have self-contradictions (like "The median cluster size is 6, 4, and 3 genes for small, medium, or large clusters, respectively"). References in text to Figure panels are frequently not matching (e.g., the sentence "We developed a new gene dependency metric, termed "perturbation score" " refers to Figure 2B, which is just a set of violin plots showing "The perturbation score profiles of the same four genes as in A."). Words missing in sentences make them hard to understand e.g., in the sentence "such that the size of the 1st and 2nd largest clusters {less than or equal to} b (b = 2.26 in our case " the word "ratio" is missing). Grammar and word usage issues additionally hamper difficulty in understanding.Perhaps most importantly, the document is really poorly structured with the definitions given after the analysis using the defined term was described, some of relevant logic found in Materials and methods section etc. While having the details of informatics analysis would be rather illuminating for the "average biology scientist", the paper still should be rewritten and streamlined to provide the better flow and eliminate the elements of sloppiness, including all points mentioned above across the main text, figures and figure legends.

We rewrote the paper to provide a better logic flow to improve the clarity. We also removed some of the terms we created, e.g., we declined the use of the term perturbation score in our revised manuscript and called combined dependency score or simply dependency score.

1) Authors' definition "We defined non- essential genes such that they are not identified as essential from the earlier analysis using the perturbation scores of the 10 , 20 and 42 most sensitive cell lines, as well as not identified as promoting its growth using the scores of the 10 , 20 and 42 least sensitive cell lines." should be justified – why was it important to exclude also genes which appear to promote growth upon knockdown from the set of non – essential genes? How many of these genes were excluded?

As for the first point, the 10, 20, and 42 most sensitive cell lines used in the analysis correspond to the 2.5^th^, 5^th^, and 10^th^ percentiles of the perturbation scores among 423 tested cell lines, respectively. We described the thresholds using “percentile” instead of “number of cell lines” in the manuscript. Moreover, we now allow users to choose from five different thresholds, namely the X-th percentiles where X = 1, 2.5, 5, 10, and 25, to define the efficacy and selectivity in shinyDepMap; small (large) X identifies genes that are required by the small (large) number of cell lines for survival.

As for the second point, genes whose loss promotes growth have the efficacy close to zero and the large positive selectivity. These 45 growth-suppressing genes (which tend to be well-known tumor suppressor genes, such as TP53, PTEN, CDKN1A) are shown as red points in Author response image 1 (overlaid in Figure 3C). Such genes are outliers in the linear regression between efficacy and selectivity, and thus removed from the regression analysis.

2) DepMap genes may have designations as "Strongly Selective" and/or "Common Essential". What is the relation between genes identified as such in this study and in DepMap? A supplementary figure (Venn diagrams) and/or table may illustrate the overlap.

The designations (common essential and strongly selective) in the DepMap portal are determined as follows: a common essential gene is a “gene which, in a large, pan-cancer screen, ranks in the top X most depleting genes in at least 90% of cell lines. X is chosen empirically using the minimum of the distribution of gene ranks in their 90^th^ percentile least depleting lines.” A strongly selective gene is defined as a “gene whose dependency is at least 100 times more likely to have been sampled from a skewed distribution than a normal distribution.”

The designations given by this definition is often inconsistent between CRISPR and shRNA and unintuitive. For example, the CCND1 gene (Author response image 2) is marked as common essential and non-essential, according to CRISPR and RNAi (*i.e.*, shRNA). In our definition, the CCND1 gene was one of the most selectively essential gene, based on the distribution of the perturbation score, which we think is a more appropriate designation based on the distributions of both CRISPR and RNAi. Their designations are not available to download in the DepMap data repository (https://depmap.org/portal/download/), so we did not incorporate the comparison between our analysis and their analysis in this paper.

**Author response image 2. respfig2:** 

3) It is not quite clear what in Figure S1B is illustrated which would be not clear from S1A.

In the original manuscript, Figure S1A shows that there are more conditions (i.e., combinations of gene targets and cell lines) on that CRISPR induced more negative scores than shRNA. Box A corresponds to conditions that only the CRISPR found essential, but shRNA did not, while Box B corresponds to conditions that shRNA found essential, but CRISPR did not. Figure S1B looks at the same thing at the gene level; we assessed whether each gene that was targeted by both CRISPR and shRNA are found essential only by CRISPR but not by shRNA, only by shRNA but not by CRISPR. It is equivalent to ask whether data points (CRISPR and shRNA scores) from targeting one gene in all the cell lines are more overrepresented within Box A or Box B using Fisher’s exact test. Figure S1B shows that more genes were overrepresented within Box A than within Box B, confirming that more genes were found essential only by CRISPR than only by shRNA.

In the revised manuscript, we rewrote this section (Figure 1) entirely. We first compared CRISPR and shRNA and discovered genes that were found commonly essential by only one of the two methods, and showed that their biases were not random but represent distinct biological functions. We next showed that the new dependency score combining the two methods were more beneficial that cover most of the commonly essential genes found by either method.

4) Selectivity is already a very vague notion, because it can arise from differences between cell types, or from the presence/absence of mutation in certain gene(s), etc. Median selectivity, used in Figure 4F, is trickier still, because different genes in the cluster may have different sources of selectivity. While Figure 4F is, indeed, a good way to illustrate a diversity of identified clusters, a word of caution would help to avoid overinterpreting the results.

We agree with this reviewer on this point. As suggested by reviewer #2, we explained whether the genes’ selectivity can be explained by the tissue-specific dependency or not, using the ADaM algorithm in the revised manuscript. In short, we see highly selectivity is due to strong lineage-dependence in some cases while in in many more cases it cannot be explained solely by that. In the latter case, mutations in certain genes might be a more significant determinant of the dependency.

5) In the application itself, Gene Clusters are very nicely presented in the Connectivity view. It would be very helpful to provide an option to save the underlying table with gene names and correlations (for example, to enable the subsequent use in Cytoscape). At the bare minimum, some way of exporting at least gene names in the cluster should be provided, apart from copying the whole table from "Clustered genes" view and then removing unused columns.

We added a link to export the connectivity graph as GraphML format, which can be viewered in Cytoscape.

6) CRISPR data seems to dominate the perturbation score. How much different the clustering based on the perturbation score is if compared to using CRISPR data alone? Could any independent measure be provided to compare the sets of clusters? E.g., distribution of connectivity scores based on protein-protein interactions, or of enrichment scores?

We thank this reviewer for the suggestion. We implemented the perturbation scores generated by mixing CRISPR and shRNA at different ratios (100:0, 80:20, 60:40, 40:60, 20:80, 0:100), so users can easily compare thee different perturbation scores. The initially-defined perturbation score was roughly the same as CRISPR:shRNA=60:40. We provided a high-level analysis to compare the performance of different perturbation scores (Figure S2).

7) In the application Gene Clusters , Correlation View, the names of the clusters can be discerned when the user zooms in. However, in the static view (Figure 1C, section 12) it should be indicated in the legend that the three vertical "lines" on the left are, in fact, the cluster labels.

We thank this reviewer for the suggestion. We added this to the label, as suggested.

Reviewer #2:Shimada at al. present a combination CRISPR and shRNA screening data from DepMap and built user-friendly browser (https://labsyspharm.shinyapps.io/depmap) for the exploration of the dataset using their metric of efficacy and selectivity.The efficacy is the "perturbation score" of the 10th most sensitive cell line for each gene, whereas selectivity is an ad-hoc measure that the authors define according to the dispersion of the efficacy between most sensitive and least sensitive cell lines. The app also allow to explore a clustering of the data computer with their algorithm ECHODOTS.The DeepMap dataset is a very important resource to identify candidate targets and cancer dependency and efficient methods that allow exploration and analysis of these data are welcome. Unfortunately the paper of Shimada does not reach neither of these aims.

We are sorry for the low writing quality of our manuscript, which might make this reviewer think our work is dissatisfying. Our primary goal of this paper is to introduce shinyDepMap as a hypothesis-generating tool for experimental, which has not been achieved by other papers analyzing the DepMap data in the past, yet incredibly important for the scientific community. We hope that the reviewer would appreciate our view.

Overall the paper is very poorly written and very difficult to follow. It starts with the presentation of the shiny app for the 2,492 essential genes that they select with their method. I understand that the aim is to allow non-programmers to access the data, the problem is the kind of analysis supporting the plots that, I think, should much more improved.

We appreciate this reviewer’s suggestion to perform and present our analysis more carefully.

The author claim to compute a novel method of the efficacy, a new methods to select genes which are selectively lethal for cell line, and also a novel clustering method. Unfortunately none of these methods is sufficiently justified by the authors of compared with the state of the art of at least the current literature. For example, they could have compared their selectivity method to the ADaM method reported in (Behan et al., 2019), moreover it is not clear why the efficacy score computed on the basis of the "perturbation score" is more robust of should be preferred with respect to other approach. Finding out how they defined the efficacy was a very difficult.

The reason why we claimed the perturbation score is more robust than the CRISPR dependency score is experimental rather than statistical. CRISPR and shRNA rely on mechanistically two distinct biological phenomena, *i.e.*, genome-editing and knockdown. While CRISPR tends to have much less off-target effect than shRNA, it still cannot be completely independent of artifacts, just like any other experimental techniques. The combination of the two methods should reduce the biases of either. As mentioned above, we also looked at genes that were found essential only by CRISPR or shRNA, but not the other method. Combined score indeed identified most of these two classes of genes.

We appreciate this reviewer for pointing us to Behan *et al.* We are convinced that the ADaM algorithm is intuitive for interpreting the dependency of lineages. We reanalyzed our data with the ADaM and identified lineage-dependent and commonly essential genes across most lineages, as was performed in the original paper. It explained to what extent the lineage-dependence can explain the selectivity we defined (Figure 4).

Regarding the novel clustering algorithm that they propose, it is very strange that the authors need to develop a novel clustering algorithms given the plethora of available methods in the literature. Why should they develop a new one and what limits of the current state of the art their algorithm should overcome it completely obscure. Moreover, from their description of the method I doubt it is really a clustering procedure at all.There is no guarantee that rule they describe "It assigns two genes into the same cluster when the distance between the genes is smaller than epsilon" you have that each point belongs to just one cluster. The authors do not specify to what data the clustering is applied. CRISPR? shRNA? Both?

We relied on the combination of t-SNE and DBSCAN, which we would say an emerging state of the art cluster algorithm. But we combined them with ensemble clustering to overcome a couple of issues. First, t-SNE was a stochastic algorithm, and some genes can be erroneously assigned to wrong clusters in one run, and second, this is a hard clustering algorithm (*i.e.*, each gene assigned to a one and only one cluster, so we cannot tell which genes are mistakenly clustered). Our rationale for implementing ECHODOTS is to compute a probability of a gene belonging to each cluster, making the hard cluster algorithm into a soft cluster algorithm. Our approach reasonably eliminates genes that are not clustered to the same cluster reproducibly. To emphasize the soft clustering feature of ECHODOTS more clearly, we introduced “probability threshold” selection bar in the shinyDepMap; genes assigned to a cluster with the high probability guarantees that the assignment of the gene to the cluster is highly consistently among multiple runs of t-SNE and DBSCAN.

We understand that the reviewer’s concern that each cluster may not always have more than one gene. We simply defined that each cluster should have more than one gene and we treated clusters containing only one gene as “noise”. We inherited this definition from original DBSCAN (M Hahsler, *et al.*, Journal of Statistical Software, 2019). There are also other cluster-with-noise algorithms. For example, the finite mixture model, also known as model-based clustering, implements this (Fraley and Raftery, Journal of Classification 20:263-286, 2003).

The data used for clustering was the perturbation scores (*i.e.*, the linear combination of CRISPR and shRNA dependency scores) of essential genes. We specified it in the previous manuscript.

Moreover, the choice of the parameters is rather arbitrary and not discussed in the paper.Overall, I suggest the authors to focus on what they think could be an improvement of the state of the art in at least one on the three areas (efficacy selectivity and clustering) and convince the reader that on what and why they are improving something.

We thank this reviewer’s suggestion to focus on the actual improvement. Our most significant contribution is the creation of shinyDepMap, which is to remove a barrier for experimental biologists to access the DepMap resource with the right questions in mind. Among the three areas pointed out by this reviewer, we emphasize that the clustering of essential genes (not all the genes) is our significant scientific contribution. The introduction of the perturbation score and the efficacy is the building blocks for it. Our clustering approach is an extension of t-SNE + DBSCAN, as mentioned above. The unrivaled quality and scale of the DepMap data helped us perform outstanding clustering compared to what was performed by others.

Regarding the shiny app itself, it is a very basic exploratory tool. A very interesting improvement that could be useful for target identification and exploitation of synthetic lethality of to include genomic information to check if the cells depending on a specific gene share some common molecular features (mutations, copy number, pathways). This could for example be done using the efficacy as an independent variable and the molecular features as predictors in the cells depending on a specific gene.

We agree with this reviewer that our shiny app is a fundamental exploratory tool. However, users are satisfied with what we provide through our personal communication, and we would like to keep it simple and straightforward for broader users. We are also concerned about the quality of information we would provide if we add biomarkers to the tool. We have performed the biomarker discovery from functional (chemical treatment) and transcriptome data in the past (*e.g.*, Shimada K, et al. *Cell Chem Biol*. 2016 and 2018), and it is technically infeasible. In the previous works, we validated the findings experimentally. However, in the analyses, we realized that the transcriptional or genetic biomarker search generates a lot of false-positive hypotheses, which can be distinguished from true-positive hypotheses only through experimental validation. Thus, we think the quality of biomarker prediction should be substantially lower than that of our clustering, and we cautiously decided not to add them to shinyDepMap because not all the users of the tool would be experienced to distinguish their difference.

Reviewer #3:In this manuscript, Shimada et al. develop a tool called shinyDepMap to enable the non-expert bioinformatician to analyze cancer dependency datasets through a user friendly interface. This tool reports gene perturbation scores, which combines essentiality scores from shRNA and CRISPR based screens for various cell lines and utilizes network analysis to identify gene cluster and essential pathways in the various cancer cell types. Overall, this is a very useful tool to the community at large, especially those with little computational background – similar to cbioportal. It will likely be widely adopted however I have a few suggestions for improvement:Overall the website is slow. I assume this is a test version, however if it is to be widely used, some effort will be needed to ensure it is able to meet the large bandwidth and demand. Ideally it should also be hosted in an independent weblink (e.g. cbioportal).

shinyDepMap is meant to be the final version for public release. We initially considered multiple options for hosting shinyDepMap, and eventually we decided to host it on a commercial server, https://www.shinyapps.io. This is the most stable and the easiest way to maintain the tool. Moreover, shinyapps.io’s dynamic architecture is scalable and is supposed to handle copes with the larger bandwidth and demand expected upon its publication without decreasing the performance. Regarding the domain name, we believe the searchability on Google is more important than the domain name. Currently, our manuscript uploaded on biorxiv is seen on one of the top hits when the term “depmap” is searched on Google, and it will likely be replaced by the paper published on a peer-reviewed journal. Thus, we do not plan to get a new domain name. When we access to shinyDepMap, it usually takes 5-20 seconds before the landing page is shown, which is considerably fast for a shiny app. If this reviewer constantly experiences longer wait time than this, we would like to figure out why the delay is caused.

It would be helpful if in addition to the interactive graphs which are very helpful, if the tool would export data in PDF/JPG for publication purposes as well as a tabular format for each cell line/perturbation score. It is sometimes difficult to identify all key genes/cell lines simply by hovering over the graphs and it is helpful to be able to obtain the output in a table format.

We tried to work on implementing the data export option, as suggested, however, the interactive graph drawing package, visGraph is not compatible with such export in a meaningful way, so we did not implement this. Instead, we let users export the graph data in text files so users can export them into Cytoscape or other graph visualization software.

It would be helpful if the perturbation scores can include a separate parameter to indicate whether the CRISPR and shRNA based essentiality scores significantly diverge. The authors explore this in some depth and report that their score is at least as good as the CRISPR screen, however it would be helpful to the user if they received some form of confidence score that indicates how well the two modalities agree/disagree. There are many reasons for discrepancy and it would be helpful to be aware of such a discrepancy when it exists.

This question essentially asks how well the perturbation score (combined dependency score) works, compared to CRISPR or shRNA scores. In the updated shinyDepMap, we let users change the mixing ratio of the two and explore how the efficacy/selectivity/cluster of the genes of interest change when changing the ratio. Moreover, we provide the overview comparing the performance of the efficacy/selectivity and the clustering depending on the mixing ratio.

While the authors perform robustness analysis when choosing their cutoff to define essential genes (10th most sensitive cell line) it is unclear still why they picked such a cutoff. Perhaps some additional explanation of the rationale would be helpful. Also in cancer types where there are more cell lines available would it be helpful to define the 10th percentile rather than the 10th cell line?

We understand this concern. We have switched our definition of threshold from the number of cell lines to the percentile. In the updated version of shinyDepMap, we let users explore X (efficacy threshold) between 1 and 25 percentiles for cutoffs.

I find the selectivity analysis summarized in Figure 3E to be perhaps the most interesting biological insight from this paper. This is important therapeutically. However the way it is phrased, selectivity compares perhaps most likely different tumor types (i.e. a drug that's toxic to breast cancer cells is less toxic to sarcoma cells for example). Therapeutically this comparison is less valuable than comparing normal to transformed/cancer cell line for each histology or on a pan cancer level. If the data exists, it would be extremely helpful to recompute selectivity comparing cancer/non-cancer cells and reproducing Figure 3E for such a comparison.

We examined whether the selectivity is due to tissue specificity or not. Unfortunately, the dependency data of the corresponding normal cells are not available in DepMap, so we could not perform the analysis as this reviewer suggested although we agree that such analysis would be strongly valuable if feasible.

Finally, it would be very helpful for the user to be able to select a subset of cell lines to perform pairwise analysis. For example if a user would like to ask what genes are essential for cell lines harboring a KRAS mutation vs cells without a KRAS mutation, having the ability to perform pairwise analysis and displaying the data for pre-selected cell lines based on a specific feature/histology would be extremely helpful and significantly expand the value of this tool. I realize that this would necessitate incorporating the mutational annotation and copy number annotation of each cell line but perhaps the effort would be well worth it if feasible.

We do agree that such analysis would be strongly beneficial, but we think that it is beyond the scope of this paper.